# Single-cell characterization of leukemic and non-leukemic immune repertoires in CD8+ T-cell large granular lymphocytic leukemia

Jani Huuhtanen [1,2,3,16], Dipabarna Bhattacharya[1,2,16], Tapio Lönnberg [4,5], Matti Kankainen [1,2], Cassandra Kerr[6], Jason Theodoropoulos[1,2,3], Hanna Rajala [1,2], Carmelo Gurnari [6], Tiina Kasanen[1,2], Till Braun[7], Antonella Teramo [8,9], Renato Zambello [8,9], Marco Herling[7,10], Fumihiro Ishida [11,12], Toru Kawakami [11,12], Marko Salmi[5,13], Thomas Loughran[14], Jaroslaw P. Maciejewski [6], Harri Lähdesmäki[3], Tiina Kelkka [1,2,17] & Satu Mustjoki [1,2,15,17✉]

T cell large granular lymphocytic leukemia (T-LGLL) is a rare lymphoproliferative disorder of mature, clonally expanded T cells, where somatic-activating *STAT3* mutations are common. Although T-LGLL has been described as a chronic T cell response to an antigen, the function of the non-leukemic immune system in this response is largely uncharacterized. Here, by utilizing single-cell RNA and T cell receptor profiling (scRNA+TCRαβ-seq), we show that irrespective of *STAT3* mutation status, T-LGLL clonotypes are more cytotoxic and exhausted than healthy reactive clonotypes. In addition, T-LGLL clonotypes show more active cell communication than reactive clones with non-leukemic immune cells via costimulatory cell–cell interactions, monocyte-secreted proinflammatory cytokines, and T-LGLL-clone-secreted IFNγ. Besides the leukemic repertoire, the non-leukemic T cell repertoire in T-LGLL is also more mature, cytotoxic, and clonally restricted than in other cancers and autoimmune disorders. Finally, 72% of the leukemic T-LGLL clonotypes share T cell receptor similarities with their non-leukemic repertoire, linking the leukemic and non-leukemic repertoires together via possible common target antigens. Our results provide a rationale to prioritize therapies that target the entire immune repertoire and not only the T-LGLL clonotype.

[1] Hematology Research Unit Helsinki, University of Helsinki and Helsinki University Hospital Comprehensive Cancer Center, Helsinki, Finland. [2] Translational Immunology Research Program and Department of Clinical Chemistry and Hematology, University of Helsinki, Helsinki, Finland. [3] Department of Computer Science, Aalto University, Espoo, Finland. [4] Turku Bioscience Centre, University of Turku and Åbo Akademi University, Turku, Finland. [5] InFlames Flagship Center, University of Turku, Turku, Finland. [6] Translational Hematology and Oncology Department, Taussig Cancer Center, Cleveland Clinic, Cleveland, OH, USA. [7] Department I of Internal Medicine, Center for Integrated Oncology (CIO), Aachen-Bonn-Cologne-Duesseldorf, University of Cologne (UoC), Cologne, Germany. [8] Department of Medicine (DIMED), Hematology and Clinical Immunology Branch, Padova University School of Medicine, Padova, Italy. [9] Veneto Institute of Molecular Medicine (VIMM), Padova, Italy. [10] Clinic of Hematology and Cellular Therapy, University of Leipzig, Leipzig, Germany. [11] Department of Biomedical Laboratory Sciences, Shinshu University School of Medicine, Matsumoto, Japan. [12] Division of Hematology, Department of Internal Medicine, Shinshu University School of Medicine, Matsumoto, Japan. [13] MediCity Research Laboratory and Institute of Biomedicine, University of Turku, Turku, Finland. [14] Division of Hematology/Oncology, Department of Medicine, UVA Cancer Center, University of Virginia, Charlottesville, VA, USA. [15] iCAN Digital Precision Medicine Flagship, Helsinki, Finland. [16]These authors contributed equally: Jani Huuhtanen, Dipabarna Bhattacharya. [17]These authors jointly supervised this work: Tiina Kelkka, Satu Mustjoki. ✉email: satu.mustjoki@helsinki.fi

T cell Large Granular Lymphocytic Leukemia (T-LGLL) is a rare lymphoproliferative disease characterized by the accumulation of abnormal, clonally restricted, and activated effector T cells in the blood, bone marrow, and spleen[1,2]. Although immune-mediated cytopenias (most frequently neutropenia, 70–80%) and autoimmune manifestations (most frequently rheumatoid arthritis [RA], 10–18%) are commonly associated with T-LGLL, it usually presents as an indolent disease that is manageable with low-dose immunosuppressive therapies[3–5].

The absence of presenting symptoms and shared morphological and phenotypic (CD3$^+$CD8$^+$CD45RA$^+$CD57$^+$) features with healthy reactive cytotoxic T cells impose challenges in the diagnosis of CD8$^+$ T-LGLL[3]. Activating somatic mutations in the STAT3 gene, the hallmark of CD8$^+$ T-LGLL, occurs in 40–50% of patients[6–11], where the majority of variants (e.g., Y640F, D661V, and D661Y) are located in the SH2 domain of STAT3. Although the clinical features of patients with mutated and wild-type STAT3 overlap, (severe) neutropenia and autoimmune manifestations are more common in mutated STAT3 cases[6,10,12–14]. Nevertheless, T-LGLL clonotypes show constant STAT3 activation irrespective of the STAT3 mutation status and are resistant to FAS/FAS-L mediated apoptosis, which can be explained by upregulated survival signaling pathways, such as MCL1, NF-κB, and PI3K/AKT[15–17].

Although chronic antigenic stimulation has been suggested to drive cytotoxic T cell lymphoproliferation in T-LGLL, little has been reported about the function of non-leukemic populations in driving or aiding T-LGLL pathogenesis. Altered B-cell activities (dyscrasias, hypergammaglobulinemia, enhanced production of immunoglobulins, including autoantibodies)[18], elevated levels of multiple cytokines (e.g., IL-15, TNF, IL-6)[19–22], and that IL-15 expression by monocytes can initiate T-LGLL in transgenic mice[19,20] suggest the possible function of non-leukemic cells in the disease. As changes in leukemia cell burden cannot be associated with therapy responses[12,23,24], and multiple symptoms can be attributed to elevated cytokine expressions[3], a holistic understanding of the total immune repertoire behind T-LGLL is an unmet need.

Here, we use single-cell RNA and TCR sequencing (scRNA +TCRαβ-seq) to separate T-LGLL clonotypes from their non-leukemic repertoire and compare them with healthy controls, other cancers, and autoimmune disorders to identify the position of T-LGLL in the intersection of cancer, autoimmune disorders, and chronic inflammation. We extend our findings with bulk-RNA-seq, TCRβ-seq, flow cytometry, serum protein profiling, and ex vivo validations. Our systems immunology analysis highlights the synergistic function of clonal and non-clonal immune repertoires in the pathogenesis of T-LGLL and suggests that future therapies should be geared toward attenuating the entire immune system and not the T-LGLL clone alone (Fig. 1a).

## Results

**T-LGLL cells show elevated cytotoxicity and exhaustion**. To gain an unsupervised view of the immune system in T-LGLL, we analyzed over 150,000 flow cytometry-sorted CD45$^+$ blood mononuclear cells (Supplementary Fig. 1a) from 11 T-LGLL samples from nine individuals and six age-matched healthy controls with scRNA+TCRαβ-seq (10X Genomics, Supplementary Data 1). After initial clustering of the entire dataset (Fig. 1b, Supplementary Fig. 2a–d), we focused on cells expressing TCR and reclustered these (Fig. 1c, Supplementary Fig. 3a–d). Despite similarities with the clonally expanded CD8$^+$ T cells (defined as at least two cells with identical TCR) from the healthy controls, the clonally expanded CD8$^+$ T cells from T-LGLL patients also had unique T-LGLL-specific characteristics, and they were overrepresented in several CD8$^+$ T cell clusters (Supplementary Fig. 3e).

As expected, samples from patients with T-LGLL were more clonal and had more expanded cells than those from healthy controls ($P < 0.01$, Mann-Whitney test) (Fig. 1d). This was invariant to the chosen threshold for expanded clones (Supplementary Fig. 4a–b). To identify transcriptomic differences between T-LGLL cells and reactive cytotoxic clonotypes, we extracted the hyperexpanded clonotypes (defined as at least 10 cells with identical TCR) from patients with T-LGLL and healthy controls and annotated them with the previously calculated clusters (Fig. 1c, e). In healthy controls, the hyperexpanded cells had preferentially CD8$^+$ effector memory (CD8$^+$ T$_{EM}$) phenotype ($P < 0.0001$, two-sided Fisher's test), whereas in T-LGLL, the hyperexpanded cells were phenotypically more heterogeneous (Fig. 1e, Supplementary Fig. 5a). In comparison with hyperexpanded reactive cells from the healthy, the top upregulated genes in T-LGLL cells included multiple cytotoxicity-associated transcripts (GZMB, PRF1, KLRB1, KLRD1), where the most significantly upregulated was NKG7, which is essential in the mobilization of cytotoxic vesicles[25] (Fig. 1f, Supplementary Data 2). Another top differentially expressed (DE) gene included common T-LGLL markers (CCL4, CCL5 [RANTES], FOS, FCGR3A [CD16], IFNG), anti-apoptotic genes (JUN, DUSP1), and genes associated with T cell exhaustion (LAG3 and TIGIT) (Fig. 1g). The DE genes translated to the top upregulated pathways in T-LGLL being cell killing, T cell activation, and response to IFNγ signaling pathways (Fig. 1h, Supplementary Data 2). In healthy controls, the top DE genes, including other cytotoxic genes (GNLY, LYZ), genes forming calprotectin (S100A8 and S100A9), and CD52, were not enriched to any immune-associated pathway (Fig. 1f, g, Supplementary Fig. 5b).

To validate the higher cytotoxicity profile in T-LGLL in comparison to reactive cells, we performed flow cytometry analysis with six T-LGLL and six healthy control samples (Supplementary Data 1). Putative T-LGLL clonotypes (CD8$^+$CD57$^+$) were confirmed to express more cytotoxic proteins (GZMA/GZMB $P < 0.01$, PRF1 $P < 0.05$, Mann-Whitney test) than the CD8$^+$CD57$^+$ T cells from healthy controls (Fig. 1i, Supplementary Fig. 6a). In addition, the CD8$^+$CD57$^+$ T-LGLL cells failed to respond well to anti-CD3/CD28/CD49 antibody-mediated TCR stimulation. Their degranulation responses (CD107a/b) were deficient ($P < 0.01$), and their cytokine production (TNF/IFNγ) in response to stimulation was diminished ($P < 0.01$) relative to the healthy CD8$^+$CD57$^+$ cells (Fig. 1j, Supplementary Fig. 6a). However, basal levels of TNF/IFNγ were higher in T-LGLL (Supplementary Fig. 6b). Since higher levels of TNF/IFNγ have been previously reported in T cells of patients with latent infections like CMV[26], we hypothesized that this phenomenon is suggestive of antigen-experienced T cell exhaustion, which is concordant with DE genes.

**Phenotypic characterization of T-LGLL clones**. The TCR-repertoires of patients with T-LGLL varied from oligoclonal to polyclonal, with the immunodominant clonotype explaining 7–50% of the total TCR repertoire (Fig. 2a). We separated the T-LGLL clonotypes from other CD8$^+$ T cells by manually curating data from (1) scTCRαβ-seq, (2) Vβ flow cytometry (presence of immunodominant Vβ family), and (3) STAT3 amplicon sequencing (matching variant allele frequency of STAT3 mutation or wild-type STAT3 with Vβ flow cytometry data, Supplementary Data 1). In total, we identified 18 T-LGLL clonotypes (nine mutated STAT3 and nine wild-type STAT3, Supplementary Data 3), and each individual patient harbored one to four T-LGLL clonotypes (Fig. 2a). Two patients had follow-up samples, and the same T-LGLL clonotypes were observed in both timepoints (four T-LGLL clonotypes in patient 1, and one in patient 2) (Fig. 2a).

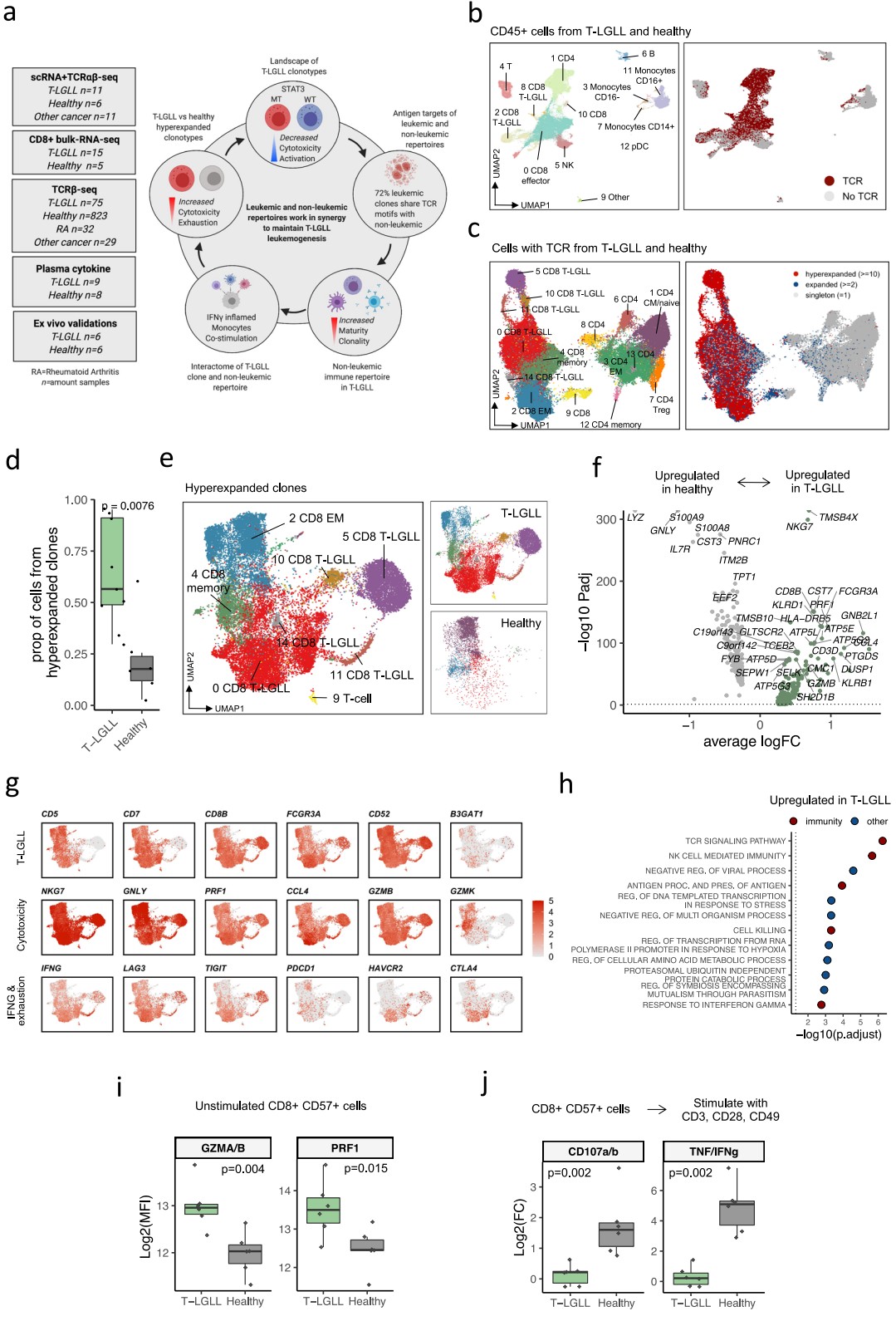

A reclustering of the cells belonging to the T-LGLL clonotypes identified seven different transcriptomic T-LGLL phenotypes (Fig. 2b–d, Supplementary Fig. 7a–b). Each T-LGLL clonotype was represented in more than one cluster (Fig. 2e). The biggest cluster, which upregulated T-LGLL markers *FCGR3A* (*CD16*) and *CCL4* and exhaustion markers *LAG3, TIGIT,* and *TOX,* was the dominant phenotype in the majority of clonotypes (13/18, 72.2%)

(Fig. 2c–e, Supplementary Data 2). Other T-LGLL phenotypes included an effector cluster with upregulated *KLRB1, IL32,* and *BATF* expression (cluster 1); a highly cytotoxic cluster (cluster 2); memory-like cluster (cluster 3); and a preferentially cytokine-producing cluster with upregulated *IFNG* and *CCL3* (cluster 4). In pathway analysis, 4/7 (57.14%) clusters (clusters 0, 3, 4, and 6) had upregulated NF-kB activation, whereas clusters 3 and 6

**Fig. 1 T-LGLL cells show elevated cytotoxicity and exhaustion as compared to healthy hyperexpanded clonotypes. a** Schematic diagram of the study, where the left panel denotes the different cohorts, and the right panel highlights the main findings. **b** Uniform Manifold Approximation and Projection (UMAP) representation of CD45$^+$ sorted cells from T-LGLL ($n = 11$) and healthy donor ($n = 6$) samples profiled with scRNA+TCRαβ-seq. Different colors indicate clusters, and cells with detected TCR are highlighted in red. **c** UMAP representation of the reclustered T cells and their phenotypes (left). Cells with detected TCR (right) were divided into singletons (TCR detected once), expanded (TCR detected ≥2 times), and hyperexpanded (TCR detected ≥10 times) clonotypes. **d** Proportion of the cells from hyperexpanded (TCR detected ≥10 times) clonotypes as compared between T-LGLL ($n = 11$) and healthy ($n = 6$). The definition of box plot visualization is stated in the Methods section Data visualization. *P*-value was calculated with two-sided Mann-Whitney test. **e** Focused UMAP of the cells with hyperexpanded TCRs from panel **c** without reclustering (left). Distribution of the cells from patients with T-LGLL and healthy controls are shown separately (right). **f** Differentially expressed genes ($P_{adj} < 0.05$, calculated with a Bonferroni corrected *t*-test) between hyperexpanded T-LGLL and healthy clonotypes. Top 30 differentially expressed genes from T-LGLL and top 10 from healthy are labelled. *X*-axis denotes the average log2 fold-change between the two conditions and *Y*-axis denotes the $P_{adj}$-value in a negative log10. Dashed line denotes $P_{adj} = 0.05$. **g** The scaled expression of the selected top differentially expressed genes highlighted using the same UMAP representation as in panel **e**. **h** Top upregulated GO-pathways ($P_{adj} < 0.05$, Benjamini-Hochberg corrected Fisher's one-sided exact test on differentially expressed genes) in T-LGLL clonotypes in comparison to hyperexpanded clonotypes from healthy controls. Colors indicate whether the pathway can be associated to immune function by manual curation. **i** Protein level expression (mean fluorescence intensity, MFI) of cytotoxic proteins (GMZA, GZMB, and PRF1) from patients with T-LGLL ($n = 6$) and healthy controls ($n = 6$) in flow cytometry cohort. *P*-values were calculated with two-sided Mann-Whitney test. **j** Protein level expression (log2 fold-change of MFI) of cytokines (TNF and IFNγ) and degranulation markers (CD107a and CD107b) between TCR stimulated and unstimulated conditions. *P*-values were calculated with two-sided Mann-Whitney test.

demonstrated a significant response to IFNγ (Fig. 2d, Supplementary Data 2). Clusters 0, 3, and 4 displayed upregulated *STAT3* transcription factor regulatory networks in SCENIC[27] analysis (Supplementary Fig. 7c–d).

**Wild-type *STAT3* T-LGLL clones are more cytotoxic than mutated clones.** Notably, *STAT3* mutated and wild-type clonotypes imputed from scTCRαβ-seq and amplicon sequencing data were partly grouped separately in the dimensionality-reduced space (Fig. 2f). To validate our manual inference of *STAT3* status, we analyzed off-target reads from the scRNA-seq data and identified 83 cells that expressed mutated *STAT3* and 200 cells that expressed wild-type *STAT3* (Fig. 2f, Supplementary Fig. 8a). *STAT3* cells expressing Y640F, S614R, and D661Y mutations were enriched in the *CD52*$^+$ memory-like cluster 3 ($P < 0.001$, two-sided Fisher's exact test), whereas the wild-type *STAT3* cells were enriched in the cytotoxic cluster 2 ($P < 0.0001$). The largest cluster (cluster 0) contained both mutated and wild-type *STAT3* cells.

DE gene and pathway analysis showed that the wild-type *STAT3* cells were more activated and displayed increased cytotoxicity compared with the mutated *STAT3* cells. The top DE genes in the wild-type *STAT3* cells included *GNLY*, *KLRG1*, and *CD5*, and the most upregulated pathways included T cell activation, upregulated TCR signaling, and response to IFNγ (Fig. 2g, h, Supplementary Data 2). Furthermore, the wild-type *STAT3* T-LGLL clonotypes also demonstrated a higher cytotoxicity score[28] ($P < 0.0001$, Mann-Whitney test) and a lower exhaustion score ($P < 0.0001$) than the mutated *STAT3* T-LGLL clonotypes (Fig. 2i, Supplementary Fig. 8b). On the contrary, the upregulated genes in the mutated *STAT3* clonotypes included genes associated with T cell survival (*JUND*, *KLF2*) and cytokine signaling (*CCL3*, *CCL4L2*, *IL2RG*), and the top upregulated pathways were associated with protein translation and response to type I interferons (IFNα, IFNβ) although none were significant after *P*-value adjustment (Supplementary Fig. 8c).

To validate the differences between mutated and wild-type *STAT3* T-LGLLs, we profiled additional patients with mutated *STAT3* ($n = 10$) and wild-type *STAT3* ($n = 5$) CD8$^+$ T-LGLL together with CD8$^+$ T cells from healthy donors ($n = 5$) with bulk-RNA-seq (Supplementary Data 1). The bulk-RNA-seq data confirmed that the wild-type *STAT3* samples were separated from the mutated *STAT3* ones in the dimensionality-reduced space by principal component 2 (PC2) which explained 15.85% of the variance (Fig. 2j). The bulk-RNA-seq data also validated the

higher cytotoxicity scores of CD8$^+$ T cells in the wild-type *STAT3* patients compared with the mutated *STAT3* patients ($P < 0.05$, Mann-Whitney test) (Fig. 2k).

**Leukemic and non-leukemic TCRs share structural similarities.** Although direct evidence is lacking, it is generally hypothesized that the T-LGLL clones originate from antigen-specific immune responses[3]. As the underlying antigen specificities of the T-LGLL clonotypes remain largely unknown, we combined previously TCRβ-seq profiled T-LGLL clonotypes[24,29] together with our samples profiled with scTCRαβ-seq ($n = 11$), TCRβ-seq from CD8$^+$ sorted samples ($n = 8$), and TCRαβs inferred from bulk-RNA-seq ($n = 15$) data (Supplementary Data 1) to form the largest described dataset of 199 T-LGLL clones from 170 patients (Supplementary Data 3). By genotyping or inferring the HLA-types[30] from scRNA-seq and bulk-RNA-seq data, we were able to determine the HLA type for 31% of the clonotypes (62/199), and 69% (43/62) were HLA-A*02 + . All T-LGLL clonotypes were restricted to individual patients, and no structural amino acid-level similarities were identified by GLIPH2[31], even when the analysis was focused only on the 43 HLA-A*02 + T-LGLL clones. This suggests the absence of shared target antigen(s) driving the clonal expansions in T-LGLL.

Next, we hypothesized that despite there being no shared antigen between patients, the non-leukemic clonotypes could target the same eliciting antigen in individual patients, which would be observed as shared TCR motifs between leukemic and non-leukemic clonotypes. Iterative GLIPH2 analysis performed on CD8$^+$ ($n = 8$) and mononuclear cell (MNC) sorted ($n = 17$) TCRβ-seq patient samples indicated that the leukemic T-LGLL clones indeed shared amino acid-level similarities with their non-leukemic repertoire in 72% of patients with T-LGLL (6/8 CD8$^+$ and 12/17 MNC-sorted samples, Fig. 3a, b, Supplementary Fig. 9, Supplementary Data 3). To avoid bias due to differences in the sequencing depth, the samples were subsampled to the same read-depth (30,000 reads per sample) before analysis. Similar results were also obtained after excluding the leukemic clone and subsampling only the non-leukemic TCRs to 30,000 reads per sample (Supplementary Fig. 10a, Supplementary Data 3). These findings denote that the majority of the leukemic T-LGLL clonotypes are likely to target the same antigen as clonotypes in their non-leukemic repertoire, and therefore, we termed them antigen-driven clonotypes.

To understand whether the antigen drive is restricted to T-LGLL, we performed a similar analysis using TCRβ-seq samples

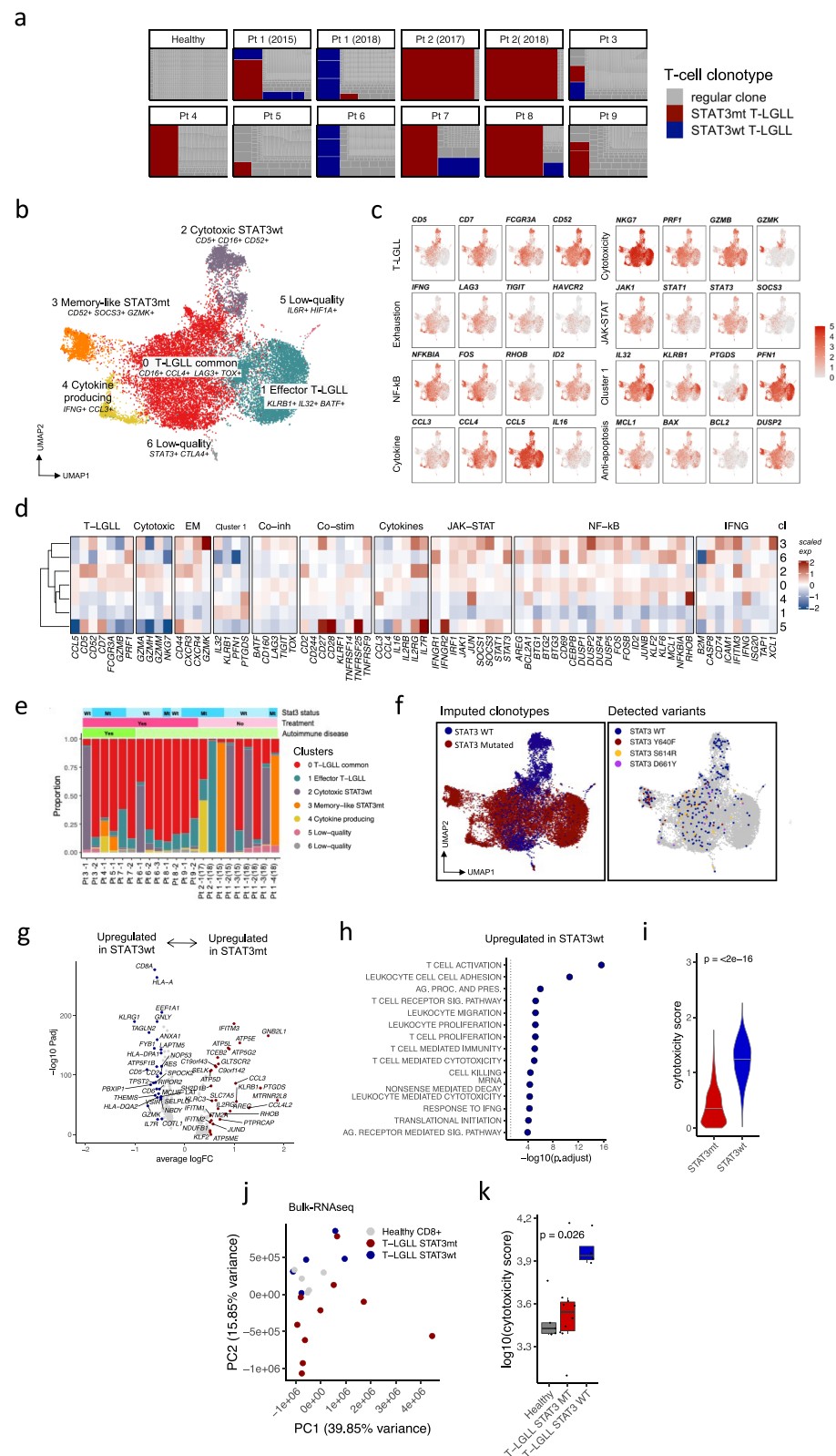

from patients with RA[32] ($n = 45$), metastatic melanoma sampled from blood[33] (SKCM, $n = 29$), and healthy controls (CD8+ sorted[32], $n = 38$; MNC-sorted $n = 785$[34]) (Supplementary Data 1) with similar subsampling. The antigen drive was the most prevalent in T-LGLL ($P < 0.05$, two-sided Fisher's exact test) and the least seen in healthy controls (Fig. 3b, Supplementary Data 4). The antigen-driven clonotypes were detected in both mutated

($n = 6$) and wild-type *STAT3* patients ($n = 6$) in equal proportions (Fig. 3c).

We next asked whether the antigens causing these polyclonal responses in T-LGLL are caused by commonly encountered antigen epitopes. Less than half (74/199, 37.19%) of the T-LGLL clonotypes were found at least once in the healthy ($n = 785$) TCRβ repertoires[34], and they explained <1% of the healthy

**Fig. 2 T-LGLL clones are phenotypically heterogeneous and wild-type *STAT3* clones are more cytotoxic than mutated *STAT3* clones. a** Clonal expansion of clonotypes in T-LGLL ($n = 11$) and one representative healthy control. Each box denotes a unique T cell clonotype in a sample as detected with scTCRαβ-seq and the size of the box corresponds to its frequency in the repertoire occupancy. **b** UMAP representation of the transcriptomes of the selected 18 T-LGLL clonotypes highlighted in panel **a** from 11 T-LGLL samples ($n = 9$ patients). **c** Scaled expression of selected differentially expressed genes between T-LGLL clusters highlighted in the same UMAP representation as in panel **b**. **d** Heatmap showing scaled expression of differentially expressed genes ($P_{adj} < 0.05$, calculated with Bonferroni corrected two-sided $t$-test) between T-LGLL phenotype clusters. Cluster (cl) numbers referring to panel **b** are marked on right side of the heatmap. **e** Proportion of the different patient-specific T-LGLL clonotypes in different clusters. Each bar represents an individual T-LGLL clone. **f** The imputed and detected *STAT3* mutation status presented in the UMAP representation. The inferred *STAT3* mutation status was obtained clonotype-wise as in panel **a** (shown in the panel on the left) and the detected *STAT3* mutation status was retrieved from the scRNA-seq data using a variant detection tool Vartrix (shown in the panel on the right). **g** Differentially expressed genes between ($P_{adj} < 0.05$, calculated with Bonferroni corrected two-sided $t$-test) the mutated and wild-type *STAT3* T-LGLL clones. Top 20 genes from each condition are labeled. *X*-axis denotes the average log2 fold-change between the two conditions and *Y*-axis denotes the $P_{adj}$-value in a negative log10 transformed scale. **h** Top upregulated GO-pathways ($P_{adj} < 0.05$, Benjamini-Hochberg corrected Fisher's one-sided exact test on differentially expressed genes) in wild-type *STAT3* in comparison to mutated *STAT3* clonotypes. **i** Cytotoxicity score of individual cells in wild-type *STAT3* clones in comparison to mutated *STAT3* clones in scRNA-seq. *P*-value was calculated with two-sided Mann-Whitney test. **j** Principal component analysis (PCA) plot from bulk-RNA-sequencing data from 10 mutated and 5 wild-type *STAT3* T-LGLL patients' and 5 healthy donors' CD8$^+$-sorted T cells. **k** Cytotoxicity score of the wild-type *STAT3* patients ($n = 5$) as compared to the *STAT3* mutated ($n = 10$) patients' scores in the bulk-RNA-seq validation cohort. *P*-value was calculated with two-sided Kruskal-Wallis test.

repertoire (Supplementary Fig. 10b, c). To control the impact of HLA genotypes, we also restricted our analysis to 43 T-LGLL clonotypes with known HLA-A*02$^+$ and searched them only among HLA-A*02$^+$ healthy donors ($n = 294$). 44.2% (19/43) of the clonotypes were found at least once, also in low frequencies (Supplementary Fig. 10d, Supplementary Data 3).

Interestingly, the antigen-driven clonotypes were more frequently observed in the healthy controls' ($n = 785$) TCRβ repertoires than the non-antigen-driven clonotypes ($P < 0.01$, Kruskal-Wallis test, Supplementary Fig. 10e), suggesting that antigen-driven clonotypes could recognize commonly encountered antigens. Therefore, we predicted the antigen specificities for T-LGLL clonotypes with a supervised machine-learning method TCRGP[35] against common viral epitopes from CMV, EBV, Influenza A, and HSV2. Only 2 of 199 (1.0 %) T-LGLL clonotypes were predicted to recognize these antigens and both clonotypes were targeting CMV pp65 epitope (Supplementary Data 3). As the TCRGP-models have been trained using data from HLA-A*02 donors, we next focused only on the 43 T-LGLL clonotypes detected in HLA-A*02+ patients. None of the T-LGLL clonotypes from either HLA-A*02+ positive ($n = 43$) or HLA-A*02 negative ($n = 19$) patients were predicted to recognize these viruses (Supplementary Data 1 and 3). The two TCRs predicted to target CMV pp65 were from patients from which HLA type was not available. Overall, these results suggest that these four viruses do not contain major driver antigens for T-LGLL.

We next studied a cohort of T-LGLL patients from which follow-up samples were available[24] ($n = 17$, 38 samples). We noted that the same patient could harbor both antigen-driven and non-driven clones, and that the clonotypes with antigen drive were larger than the non-driven clonotypes during follow-up ($P < 0.05$, Mann-Whitney test, Fig. 3d, Supplementary Fig. 10f) suggesting that antigen drive can potentially provide a growth advantage for the clones.

To further understand the evolution of the antigen-driven and non-driven clones, we studied samples from a patient with both types of clones (patient 1). The analysis was performed using three peripheral blood samples collected seven years apart (2011–2018). Although the patient had no treatment for her T-LGLL disease, the dominant *STAT3* mutated clone (78% → 10%) which was not antigen-driven, was replaced by an antigen-driven *STAT3* wild-type clone (5% → 24%) carrying a different TCRαβ (Fig. 3e, f, Supplementary Fig. 11a–e). The non-antigen-driven *STAT3* mutated and the antigen-driven *STAT3* wild-type clones were phenotypically different and possibly two different maturation end-points as suggested by trajectory analysis with Slingshot[36] (Fig. 3f). The expanding, antigen-driven wild-type *STAT3* clone was more

cytotoxic than the shrinking *STAT3* mutated clone (Fig. 3f, Supplementary Fig. 11a–e). The top DE genes in the expanded clone included *GZMH*, *GNLY*, and *FCGR3A* (*CD16*) and the clone had a higher cytotoxicity score than the shrinking one ($P < 0.0001$, Mann-Whitney test, Fig. 3f, Supplementary Fig. 11d, Supplementary Data 2). Conversely, the *STAT3* mutated shrinking clone presented an attenuated CD8$^+$ T$_{EM}$ phenotype marked by the expression of *GZMK* and upregulated *SOCS1* and *SOCS3*, which are known to inhibit JAK-STAT signaling[19].

To conclude, our results suggest that a majority, but not all, T-LGLL clonotypes could originate from an initial polyclonal antigen-response. Also, the antigen-driven clonotypes are larger and often display a more cytotoxic phenotype than the non-driven T-LGLL clonotypes.

**In T-LGLL non-leukemic T cell populations are mature and clonal.** After finding the connection that clonal and non-clonal immune cell repertoires are possibly connected via antigen preferences, we sought to investigate the phenotypes of non-leukemic immune cell populations in T-LGLL in detail. As the presence of T-LGLL clonotypes biases the immune repertoire, we removed the clonally expanded T-LGLL cells from scRNA+TCRαβ-seq data and compared it with similar data from solid cancers[37] ($n = 3$), hematologic cancers[38] ($n = 8$), and healthy controls ($n = 6$) (Supplementary Data 1). After clustering (Fig. 4a, Supplementary Fig. 12a–c), we observed that, in comparison with the other cancers, the proportion of conventional dendritic cells (cDCs) ($P < 0.01$ and $P_{adj} = 0.052$, Benjamini-Hochberg corrected Mann-Whitney test) and naïve B-cells ($P < 0.05$ and $P_{adj} = 0.16$) were reduced, and the proportion of mature CD4$^+$ T$_{EM}$-cells (clusters 2 and 13, both $P < 0.01$ and $P_{adj} = 0.052$) was increased in T-LGLL (Fig. 4b, Supplementary Fig. 13a–b). When including only patients with hematologic cancers ($n = 8$), CD4$^+$ T$_{EM}$ cells were still markedly elevated in T-LGLL (Fig. 4b). Similar results were obtained when compared to healthy controls ($n = 6$, Supplementary Fig. 13c, d).

A patient cohort profiled with flow cytometry validated that patients with T-LGLL had a significantly higher percentage of mature terminally differentiated, antigen-experienced CD4$^+$CD57$^+$ T cells[39,40] when compared with the healthy controls ($P < 0.05$, Mann-Whitney test, Fig. 4c, Supplementary Fig. 13e–g). To support the maturity of the CD4$^+$ T cells in T-LGLL, we noted that the proliferative capacity of the total CD4$^+$ T cell compartment, measured as carboxyfluorescein succinimidyl ester (CFSE) dilution upon TCR ligation and TLR stimulation, was reduced in T-LGLL compared with the healthy controls ($P < 0.05$, Fig. 4d, Supplementary Fig. 13f).

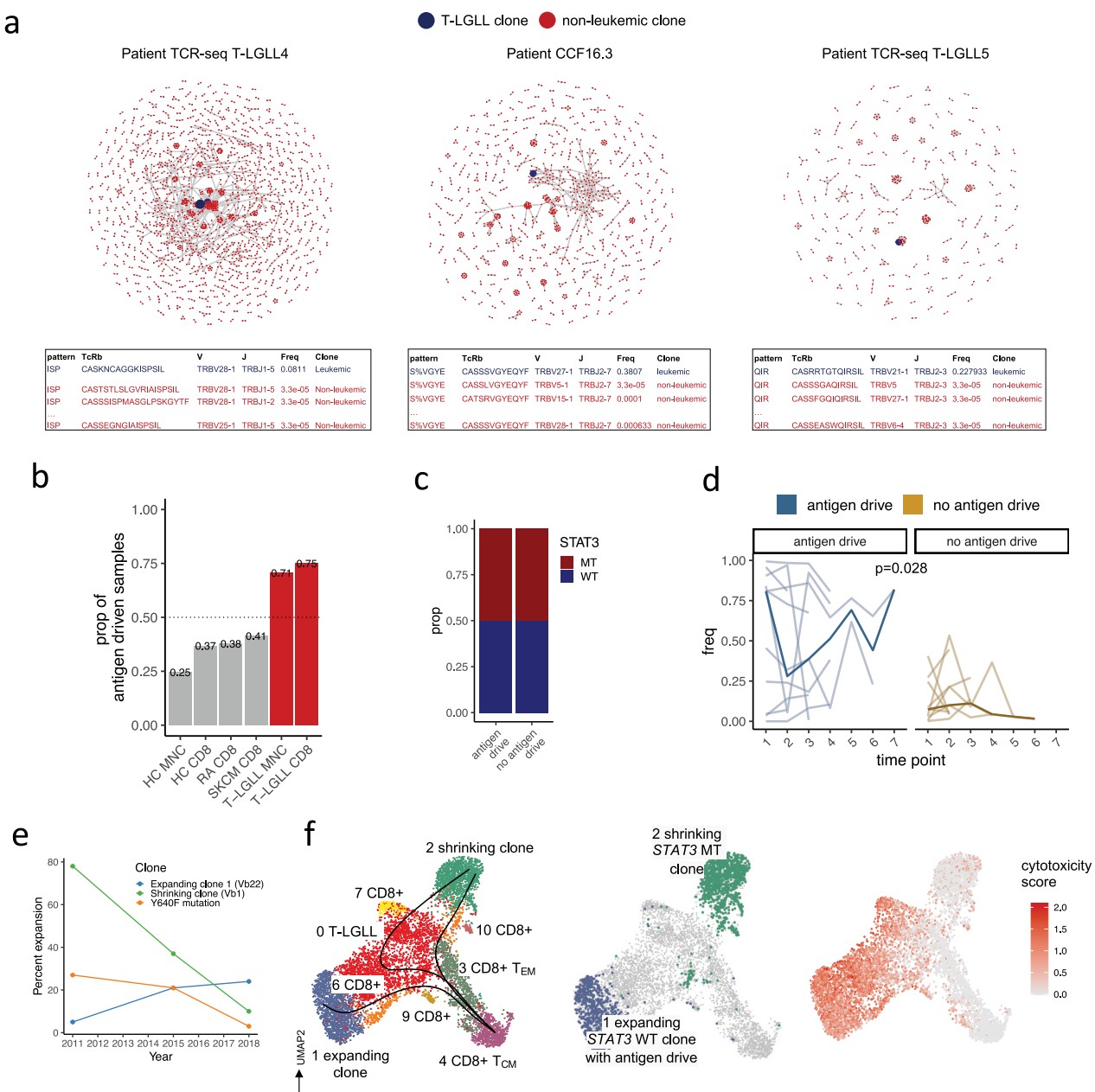

**Fig. 3 T-LGLL clonotypes' TCRs share structural similarities with their non-leukemic counterpart. a** Network plots showing antigen-driven clonotypes from three selected patients with T-LGLL. Antigen drive denotes that the T-LGLL clone shares amino acid-level similarities with its non-leukemic repertoire. Each dot (a node) is a TCR clonotype, and clonotypes with shared amino acid-level similarities are connected by a line (an edge). The T-LGLL clones are highlighted with blue and the non-leukemic with red. Below each network plot are shown parts of the GLIPH2 results for the individual patient with the same color coding on the TCRs. Additional T-LGLL cases are shown in the Supplementary Fig. 9. **b** The presence of antigen drive (i.e., whether the largest clonotypes have shared amino acid-level similarities with the rest of the TCR repertoire) in T-LGLL (mononuclear cell [MNC]-sorted $n = 17$, CD8+-sorted $n = 8$), metastatic melanoma sampled from blood (SKCM, $n = 29$), rheumatoid arthritis (RA, $n = 32$), and healthy controls (HC, MNC-sorted $n = 785$, CD8+-sorted $n = 38$). T-LGLL patients have more antigen-driven cases than the rest of the conditions ($P < 0.05$, Fisher's one-sided exact test). All samples were subsampled to the same read-depth (30,000 reads per sample). Results where the T-LGLL clone was excluded before downsampling and in which the subsampling was only done for the non-leukemic library are shown in the Supplementary Fig. 10a. **c** The proportion of mutated ($n = 6$) and wild-type STAT3 patients ($n = 6$) where antigen-driven or no antigen-driven clonotypes were detected in the MNC-cohort. **d** The evolution of antigen-driven and non-antigen-driven T-LGLL clonotypes in multiple timepoints. Individual lines correspond to individual T-LGLL clonotypes while the bolded line shows the median. P-value was calculated with two-sided Mann-Whitney test. **e** Flow cytometry analysis of Vβ repertoires and variant allele frequency (VAF) of STAT3 Y640F clone (located in the shrinking Vβ1 clone) are used to demonstrate T-LGLL clonal dynamics (clonal drift) in patient 1. **f** UMAP representation of the CD8+ T cells from patient 1 from two different timepoints. The left panel highlights the different clusters, and the superimposed lines correspond to predicted maturation trajectories (pseudotime) calculated with a pseudotime algorithm Slingshot. The middle panel illustrates the expanding and shrinking clones. The panel on the right highlights the previously defined cytotoxicity score in T-LGLL clonotypes.

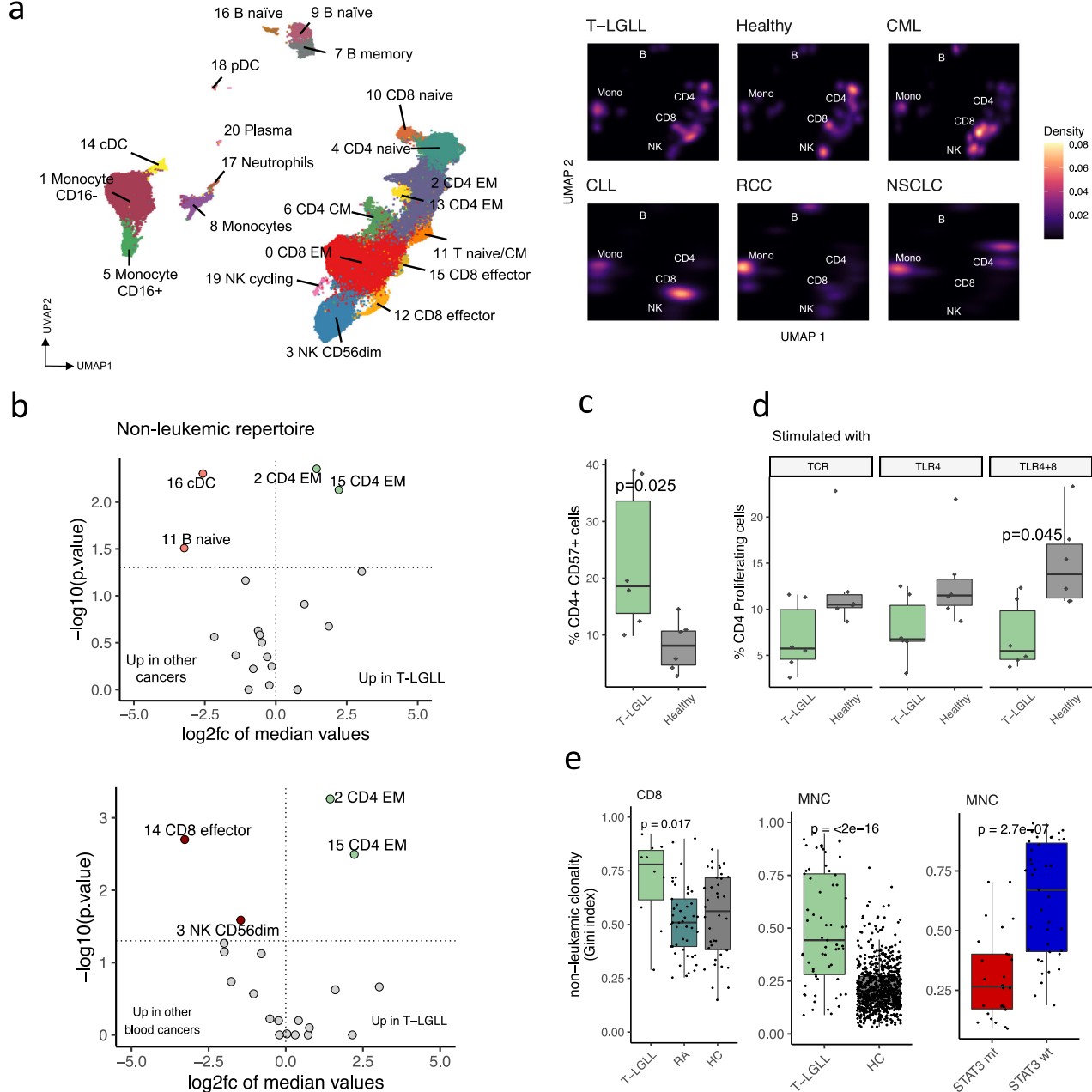

**Fig. 4 Non-LGLL T cell populations are more mature, clonal, and cytotoxic in T-LGLL compared with T cells of healthy controls and patients with other cancers. a** UMAP representations of non-leukemic CD45[+] sorted cells from 11 T-LGLL, 6 healthy, 4 CML, 4 CLL, 2 RCC, and 1 NSCLC samples profiled from peripheral blood with 10X technologies, where different colors indicate clusters. Density estimates showing the overlapping dots for each cohort are presented on the right. **b** Differentially abundant non-leukemic clusters (from panel **a**) between patients with T-LGLL ($n = 9$) and patients with other cancers ($n = 11$, upper panel) and T-LGLL and with blood cancers ($n = 8$, lower panel). The horizontal line indicates $P = 0.05$, as calculated with two-sided Mann-Whitney test. **c** Percentage of mature effector memory CD4[+]CD57[+] cells out of CD4[+] T cells in T-LGLL ($n = 6$) as compared with healthy controls ($n = 6$) in the flow cytometry cohort. $P$-values were calculated with two-sided Mann-Whitney test. **d** Percentage of proliferating CD4[+] T cells out of CD4[+] T cells measured with CFSE dilution after stimulation with either TCR ligation or TLR stimulation in patients with T-LGLL compared with healthy controls in the flow cytometry cohort. $P$-values were calculated with two-sided Mann-Whitney test. **e** Left: Clonality index (Gini, higher denotes more clonal) in CD8[+] sorted populations from T-LGLL ($n = 10$), rheumatoid arthritis (RA) ($n = 32$), and healthy controls ($n = 38$) profiled with TCRβ-seq. Middle: Clonality index in non-leukemic TCR-repertoires of mononuclear cell (MNC) samples in patients with T-LGLL ($n = 38$) and healthy controls ($n = 785$). Right: Clonality index in non-leukemic TCR-repertoires of MNC samples in *STAT3* mutated (mt) ($n = 26$) and wild-type (wt) ($n = 39$) patients. $P$-values were calculated with two-sided Mann-Whitney test.

Besides increasing T cell maturity, antigen-driven processes increase T cell repertoire clonality. Therefore, we compared the clonality of the non-leukemic CD8[+] T cells in T-LGLL to CD8[+] sorted healthy and RA samples[24,32] and found that the non-leukemic CD8[+] T cells in patients with T-LGLL had a more restricted TCR repertoire than patients with RA ($P < 0.01$, Mann-Whitney test) and healthy controls ($P < 0.05$), latter of which was validated in the MNC-cohort ($P < 0.0001$, Fig. 4e). In addition, the non-leukemic repertoires of wild-type *STAT3* patients were more clonal than those of the mutated *STAT3* patients ($P < 0.0001$) (Fig. 4e).

**IFNγ drives activation of the non-leukemic immune cell repertoire**. Besides changes in cell abundances, scRNA-seq also showed the activation of different non-leukemic subsets in T-LGLL in comparison with patients with other cancers and healthy controls. For example, the expression of different cytokines (*CCL2/3/4/5*), co-stimulatory genes (*CD27, TNFRSF4 [HVEM], TNFRSF14 [OX40], TNFRSF25 [DR3]*), and IFNγ response genes (e.g., *B2M, TAP1*, HLA molecules) were upregulated in different non-leukemic NK-cells, monocytes, and B-cell clusters in comparison with healthy controls (Fig. 5a), other cancers (Supplementary Fig. 14a), and patients with blood cancers (Supplementary Fig 14b, Supplementary Data 2). Notably, the expression of different cytotoxic genes (*GZMA/B/H, PRF1, NKG7*) was upregulated in non-leukemic CD8$^+$, CD4$^+$, and NK-cell clusters. The phenotype of the non-leukemic T cells was validated in the flow cytometry cohort, where CD8$^+$CD57$^-$ and total CD4$^+$ populations in T-LGLL expressed higher levels of GZMA/B ($P < 0.01$ for CD8$^+$CD57$^-$ and CD4$^+$, Mann-Whitney test) and PRF1 ($P < 0.01$ for CD8$^+$CD57$^-$ and CD4$^+$) than healthy (Fig. 5b).

To understand the pathways driving immune activation in T-LGLL, we performed pathway analysis among patients with T-LGLL, patients with other cancers, and healthy subsets. The most upregulated pathways in T-LGLL included IFNγ-response (upregulated in 12/16 subsets vs. other cancers, 16/17 vs. healthy, 9/15 vs other blood cancers), IFNα-response (9/16 vs. other cancers, 15/17 vs. healthy, 8/15 vs other blood cancers), and NFκB (11/16 vs. other cancers, 0/17 vs. healthy, 4/15 vs other blood cancers) pathways (Fig. 5c, Supplementary Fig. 14c, d).

We focused on the IFNγ response, as it was among the most upregulated pathway in all comparisons and quantified its effect by calculating an IFNγ response module score[41] in all immune subsets in individual patients. The strongest IFNγ response was seen in different myeloid subsets (CD16$^+$ monocytes, CD16$^-$ monocytes, cDCs), NK-cells, and CD8$^+$ T$_{EM}$ cells (Fig. 5d). In unsupervised clustering, the samples were split into two groups, high IFNγ and low IFNγ, based on their IFNγ-response scores. The samples from the T-LGLL group were enriched to the high-IFNγ group ($P < 0.05$, Fisher's one-sided exact test), confirming that IFNγ response is more strongly activated in T-LGLL than in other cancers. A similar analysis with the NF-κB pathway did not show enrichment of T-LGLL samples (Supplementary Fig. 14e). Interestingly, T-LGLL cells expressed higher amounts of *IFNG* than non-leukemic cells ($P < 0.0001$, Mann-Whitney), where the highest expression was seen in cytokine-secreting T-LGLL cluster 4 (Fig. 5e).

**T-LGLL clones show an elevated amount of predicted cell–cell interactions**. To further understand the function of cytokines as mediators of immune responses between the leukemic and non-leukemic compartments in T-LGLL, we reanalyzed the plasma cytokine profiles of nine T-LGLL patients and eight healthy controls[42] (Supplementary Data 1). In patients with T-LGLL, multiple cytokines, including IFNγ-inducible cytokines (CXCL10, CXCL11), JAK-STAT pathway-activating cytokines (IL-6, IL-10, IL-15RA), and inflammatory chemokines (CCL3, CCL4, MCP1), were elevated (Fig. 6a). However, in line with previous publications[20,21], IFNγ levels were not elevated in T-LGLL. The incorporation of plasma cytokine data with scRNA-seq showed that the majority (11/17, 64.7%) of upregulated cytokines were expressed predominantly by monocytes or cDCs (e.g., *CCL2/3/7, CXCL10/11, IL15RA*) instead of T-LGLL clonotypes (Fig. 6b, Supplementary Fig. 15a). Conversely, 6 of 17 upregulated cytokines (e.g., *CCL4, TNFRSF9 [CD137], TNFRSF14 [HVEM]*) and *IFNG* were preferentially expressed by T-LGLL clones.

In addition to being the most important cytokine producers, monocytes were also the most transcriptionally altered

subpopulations between T-LGLL and other conditions in the DE gene analysis (first, CD16$^+$ monocytes; fourth, CD16$^-$ monocytes) (Supplementary Fig. 15b–g). Flow cytometry analysis also confirmed that although the total number of monocytes was reduced ($P < 0.05$, Mann-Whitney test), the distribution of different monocyte subsets was altered, and T-LGLL patients had a bigger proportion of CD16$^+$ cells ($P < 0.05$, Supplementary Fig. 16a, b) out of the CD14$^+$ monocytes. The upregulated DE genes in monocyte populations included multiple HLA molecules and classical scavenging receptors (e.g., *CLEC10A, CD44, CLEC2B, CLEC9A, MRC1*), translating into upregulated HLA class II[28] ($P < 0.0001$, Kruskal-Wallis test) and scavenging scores ($P < 0.0001$, Fig. 6c, Supplementary Fig. 16c, Supplementary Data 2). To analyze the antigen-presenting function of the monocytes, we incubated blood MNCs with fluorescent microspheres and found that the proportions of bead-adhering CD14$^+$CD16$^+$ and CD14$^{dim}$CD16$^+$ monocytes were increased in T-LGLL compared with healthy controls ($P < 0.05$ Mann-Whitney test, Fig. 6d, Supplementary Fig. 16d), which may indicate higher scavenging potential.

Next, we calculated ligand–receptor interactions with CellPhoneDB[43] between T-LGLL clonotypes and other immune cells and compared that to the interactome of hyperexpanded clonotypes from healthy controls. The interactome analysis implicated an increased number of interactions between T-LGLL clonotypes and other immune cells in comparison with healthy hyperexpanded clonotypes (Fig. 6e). The majority of the differences could be tracked to T-LGLL–monocyte interactions, and many of the predicted interactions could be attributed as co-stimulatory (e.g., *CD2–CD58, CD48–CD244, CLEC2B–KLRF1, TNFSF14–TNFRSF14*); while only a few interactions were inhibitory (e.g., *LGALS9–HAVCR2*) (Fig. 6f). Based on the number of ligand–receptor interactions, T-LGLL clonotypes formed three clusters: (1) strongly interacting (the highest number of interactions), (2) interacting, and (3) immune independent (the lowest number of interactions), which was also evident in the focused clustering of T-LGLL clonotypes (Fig. 6g). Immune independent *STAT3* mutated T-LGLL clonotype from patient 2 had the lowest number of interactions, and during the 1-year follow-up, the size and the phenotype of this clone were stable (Supplementary Fig. 17a–e).

## Discussion

The asset of scRNA+TCRαβ-seq in this study is its accuracy in identifying the TCR-sequence-restricted clonal expansions. In other non-T cell-malignancies, similar cell-specific markers are rarely available or require simultaneous DNA sequencing to define clonal cells. With scRNA+TCRαβ-seq, we were able to perform detailed and precise characterizations of the expanded T-LGLL clones from the oligo- and polyclonal CD8$^+$ T cell repertoires and show evidence of a strong antigen-driven immune response that shapes the entire immune cell repertoire in T-LGLL.

T-LGLL clonotypes are known to overexpress cytotoxic and T cell activation-associated genes in comparison with their healthy reactive counterparts[44,45]. Here, we also found exhaustion-associated genes, such as *LAG3* and *TIGIT*, among the most upregulated genes between T-LGLL and hyperexpanded cells in healthy controls, but not the previously found *PDCD1* (*PD1*) and *HAVCR2* (*TIM-3*)[44,45]. Our in vitro validation suggested that TCR ligation fails to trigger normal degranulation and cytokine production in the T-LGLL clonotypes. These findings may partly explain why T-LGLL usually presents only with moderate lymphocytosis and rarely develops into a more aggressive (pro-liferative) disease despite highly activating *STAT3* and *STAT5B* mutations[46,47].

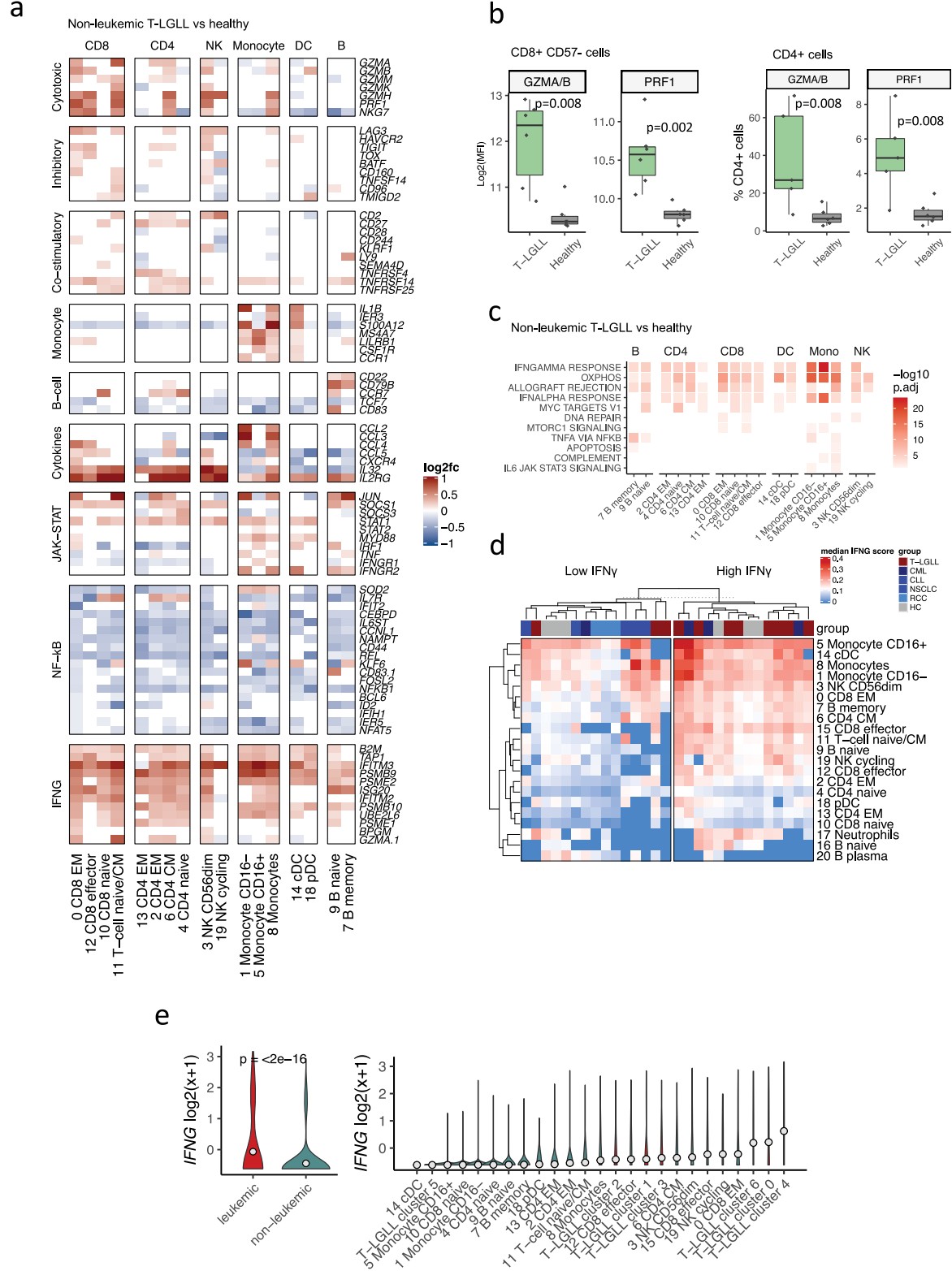

Our data demonstrate inter and intrapatient heterogeneity, where T-LGLL clones with the same TCR rearrangement can harbor multiple phenotypes. Importantly, the $CD16^+$ $CCL4^+$ $LAG3^+$ $TOX^+$ phenotype was identified as the dominant phenotype in most clonotypes (13/18, 72.2%) and was seen in patients with either mutated or wild-type *STAT3*, further unifying these diseases besides the noted shared JAK-STAT activity[19]. As this phenotype differs significantly from the effector memory phenotype of hyperexpanded $CD8^+$ T cells in healthy controls, it could aid the diagnostic process, particularly in the distinction of wild-type *STAT3* cases from reactive processes. However, the finding that wild-type *STAT3* clonotypes have higher T cell activity, cytotoxicity and non-leukemic clonality than those with mutated *STAT3*, which was not seen in a previous publication

**Fig. 5 IFNγ secretion by T-LGLL clonotypes drives the activation of the non-leukemic immune cell repertoire. a** Expression of selected differentially expressed genes ($P_{adj} < 0.05$, calculated with Bonferroni corrected two-sided $t$-test) grouped by their functional pathways between the non-leukemic CD45[+] sorted cells from patients with T-LGLL ($n = 9$) and healthy controls ($n = 6$). Values are presented as log2 fold-change (log2fc). **b** Left: Protein level expression (MFI mean fluorescence intensity) of cytotoxic proteins GZMA/B and PRF1 in CD8[+]CD57[−] cells in T-LGLL patients ($n = 6$) and healthy controls ($n = 6$). Right: The proportion of GZMA/B and PRF1[+] CD4[+] cells in the flow cytometry cohort. $P$-values calculated with two-sided Mann-Whitney test. **c** Upregulated HALLMARK-category pathways ($P_{adj} < 0.05$, Benjamini-Hochberg corrected Fisher's one-sided exact test on differentially expressed genes) in non-leukemic cells from T-LGLL ($n = 9$) in comparison with healthy ($n = 6$). **d** Median expression of the IFNγ response module score in different immune subsets in patients with T-LGLL ($n = 9$), healthy controls ($n = 6$), and patients with other cancers ($n = 11$). The T-LGLL samples were enriched in the IFNγ high cluster ($P < 0.05$, Fisher's one-sided exact test). Clustering was performed with Ward's linkage. **e** Left: Scaled expression of *IFNG* in leukemic (red) and non-leukemic (green) populations. $P$-value was calculated with two-sided Mann-Whitney test. Right: Scaled expression of *IFNG* in different leukemic (red) and non-leukemic populations (green). Cluster numbers refer to Fig. 2b (leukemic clusters) and Fig. 4a (non-leukemic clusters).

with flow cytometry[14], proposes that mutated *STAT3* T-LGLL, wild-type *STAT3* T-LGLL, and reactive processes arise from different pathogeneses.

The eliciting antigen in T-LGLL has remained elusive. We analyzed TCRs in both unsupervised and supervised manners with the current best-practice bioinformatics tools[31,35,48] but found no evidence of common putative, known or unknown antigens, even in individual patients. The obvious limitation in these analyses is that they were done independently of HLA-genotype or involved only T-LGLL clones from patients with HLA-A*02+ background. Unfortunately, HLA information was not available from all patients that were included in the previously published T-LGLL TCR datasets[24]. Supervised TCRGP tool has shown to outperform other similar methods[35], when the genotype of the analyzed TCR repertoire is known. However, as the training data of epitope-specific TCRs is limited, it is probable that the existing TCR analysis tools do not capture the full heterogeneity of antigen-specific repertoire, resulting in false negatives even in the cases of HLA-A*02+ patients. Nevertheless, our results imply that the common denominator underlying T-LGLL patients is perhaps not the antigen, but rather the environmental, genetic, and/or immunological factors that support the expansion and persistence of T-LGLL clonotypes. These results are in accordance with Gao et al.[49], who profiled alemtuzumab treated T-LGLL patients with scRNA +TCRαβ-seq, and no shared T-LGLL clonotypes or T-LGLL clonotypes targeting known antigens were observed.

Our results do not, however, contradict that T-LGLL is driven by an abnormal response to an antigen. On the contrary, in an analysis that is invariant to HLA genotypes, we observed that over half (72%) of the T-LGLL clonotype TCRs share structural similarities with TCRs from the same patients' non-leukemic repertoires. Our results from the antigen drive support the view that the antigen response in T-LGLL is poly- or oligoclonal, rather than monoclonal. Our results are in line with the previous data suggesting that *STAT3* mutation follows the initial clonal expansion and is an event that solidifies the clonal dominance[3]. The antigen-driven clonotypes in T-LGLL patients were larger, and they could occur concomitantly with non-antigen-driven clones. Interestingly, in one patient with follow-up samples, the mutated *STAT3* clone was replaced by a more cytotoxic wild-type *STAT3* clone. Further, the non-leukemic CD8[+] and CD4[+] T cell repertoires in T-LGLL were more mature, cytotoxic, and clonally restricted than in other cancers, in RA, and in healthy controls, suggesting the strong immune-editing capacity of a driving antigen. The advent of high-throughput epitope-MHC-TCR-screening tools[50] and their use in T-LGLL will provide invaluable information about the antigen-specific response in general.

Also other findings, besides non-leukemic CD8[+] and CD4[+] T cells, further support the idea of an aberrant oligoclonal immune response against a patient-specific antigen as a disease-inducing and evolution-driving trigger in T-LGLL. We noted increased co-stimulatory cell–cell interactions between T-LGLL clonotypes and

monocytes and enhanced antigen-presenting cell function of monocytes. The immunological factor driving these differences was a response to IFNγ and it was the most evident in monocyte populations. The *IFNG* was preferentially expressed by T-LGLL clonotypes, and not by monocytes, linking the leukemic and non-leukemic repertoires into a vicious cycle. With only incidental cases of clonal drift seen in our data, we cannot pinpoint whether the non-leukemic immune repertoire caused the transformation of a T cell clonotype to a T-LGLL clone or vice versa, which needs to be addressed in future studies.

Current therapies in T-LGLL, including corticosteroids, methotrexate, and cyclosporine A, offer unsatisfactory results, as over half of patients eventually relapse[18], posing a need for combined or sequential therapies. Current salvage therapies include T cell depleting anti-CD52 (alemtuzumab) and anti-CD3 (anti-thymocyte globulin) regimens[3,51,52]. These approaches also target non-T-LGLL clones which could explain why TCR repertoire does not diversify after alemtuzumab treatment[49] and why treatment responses do not correlate with the *STAT3* mutation status or clonal burden[23]. Moreover, treatments that attenuate the entire immune system have shown encouraging results, both as first-line (cyclophosphamide, >70% response rate)[53] and salvage therapies (tofacitinib, a JAK3 inhibitor >60% response rate)[54].

In conclusion, our study highlights how the entire immune cell repertoire, including hyperexpanded CD8[+] T-LGLL cells, non-leukemic CD8[+] cells, CD4[+] cells, and monocytes, contribute to the CD8[+] T-LGLL disease phenotype. An aberrant antigen-driven immune response shapes the repertoire and maintains the persistence of the hyperexpanded T-LGLL clonotypes. Our results imply that future therapies should not only target the T-LGLL clonotypes but also other immune cell types and their interactions to transform the outcome of patients with T-LGLL.

## Methods

**T-LGLL patients.** Samples from T-LGLL patients were collected at the Helsinki University Hospital Comprehensive Cancer Center (Finland), Cleveland Clinic (USA), University Clinic of Cologne (Germany), University Hospital of Padova (Italy), and Shinshu University School of Medicine (Japan). Patient details can be seen in Supplementary Data 1. The study was approved by local ethical committees. Written informed consent was received from all patients and the study was conducted in accordance with the Declaration of Helsinki, and no donors were compensated for their efforts. Mononuclear cells (MNCs) were separated from peripheral blood (PB) using Ficoll-Paque PLUS (GE Healthcare). The following institutes ethically approved the protocol: Hematology Research Unit Helsinki, University of Helsinki and Helsinki University Hospital Comprehensive Cancer Center, Helsinki, Finland; Translational Hematology and Oncology Department, Taussig Cancer Center, Cleveland Clinic, Cleveland, OH, USA; Department I of Internal Medicine, Center for Integrated Oncology (CIO), Aachen-Bonn-Cologne-Duesseldorf, University of Cologne (UoC), Cologne, Germany; Clinic of Hematology and Cellular Therapy, University of Leipzig, Leipzig, Germany; Department of Medicine (DIMED), Hematology and Clinical Immunology Branch, Padova University School of Medicine, Italy; Department of Biomedical Laboratory Sciences, Shinshu University School of Medicine, Matsumoto, Japan; Division of Hematology, Department of Internal Medicine, Shinshu University School of Medicine, Matsumoto, Japan; and Division of Hematology/Oncology, Department of Medicine, UVA Cancer Center, University of Virginia, Charlottesville, VA, USA.

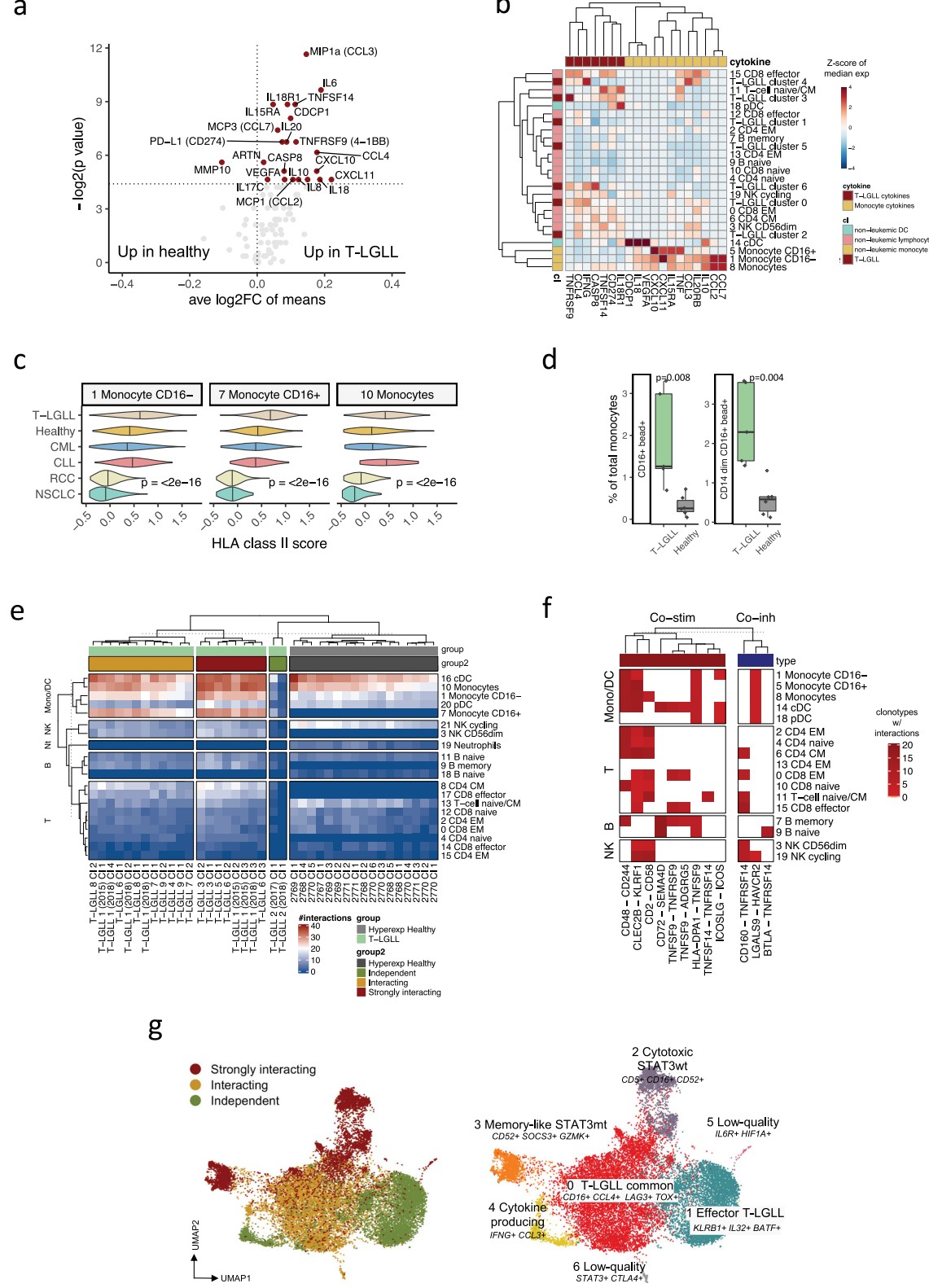

**Flow cytometry and Vβ staining.** T cell receptor (TCR) Vβ families were analyzed from T-LGL leukemia patients' whole blood or PB MNC samples by combining 0.5 ul of anti-CD3 APC (Clone: SK7, Cat#: 557851, Lot#: 8037645, BD Biosciences), 4 ul of anti-CD4 PerCP (Clone: SK3, Cat#: 345770, Lot#: 6281605, BD Biosciences), 0.8 ul of anti-CD8 PE-CY7 (Clone: SK1, Cat#: 345774, Lot#: 82152, BD Biosciences) antibodies with the panel of TCR Vβ antibodies (10ul/sample) corresponding to 24 members of variable regions of the TCR β chain (~70% coverage of normal human TCR Vβ repertoire) (IOTest Beta Mark TCR Vbeta Repertoire Kit, Cat#: IM3497, Lot#: 66, Beckman Coulter). Staining was done in 100 ul of whole blood and stained samples were analyzed with FACSVerse (BD Biosciences) and FlowJo software (Version 10.4.2, Becton Dickinson).

**Amplicon sequencing.** To detect *STAT3* mutations, locus-specific primers were designed covering the Src homology 2 (SH2) domain of STAT3 (exons 19–24) as reported previously[6]. The list of primers used in this study is provided in Supplementary Data 1. Illumina HiSeq System was used as described previously[55]. Briefly, 2 step PCR protocol (Illumina) was used with coverage of over 100,000× and

**Fig. 6 T-LGLL clonotypes have increased amounts of predicted cell–cell interactions, especially with monocytes. a** Differentially expressed (unadjusted two-sided Mann-Whitney test) plasma cytokines between patients with T-LGLL ($n = 9$) and healthy controls ($n = 8$), where cytokines $P < 0.05$ (horizontal line) are labeled. **b** Median expression of differentially expressed cytokines in the scRNA-seq data in non-leukemic and leukemic immune cell subsets in the patients with T-LGLL (cluster numbers refer to Fig. 4a). Heatmap clustering was performed with Ward's linkage. Values are scaled for each column. **c** HLA class II module score in T-LGLL in different monocyte clusters (as seen in Fig. 4a) in comparison with healthy controls and other disease cohorts. $P$-values were calculated with two-sided Kruskal-Wallis test. **d** Proportion of bead-adhering (fluorescent microspheres) CD16+ and CD16+ CD14dim monocytes in patients with T-LGLL ($n = 6$) in comparison to healthy controls ($n = 6$). $P$-values were calculated with Mann-Whitney test. **e** Number of significant ligand–receptor interactions ($P < 0.05$, CellPhoneDB permutation test) between T-LGLL clonotypes (as shown in Fig. 2e) or the top expanded hyperexpanded clonotypes (>50 TCRs) from healthy controls and different non-leukemic immune cell subpopulations, calculated CellPhoneDB. Clustering was performed with Ward's linkage. Color scale from blue to red marks the number of predicted interactions between cell types. **f** Number of significant ligand–receptor interactions of T-LGLL clonotypes with different immune subpopulation. Shown receptor–ligand pairs are statistically significant interactions that have been attributed as co-stimulatory or inhibitory. The color indicates the number of T-LGLL clonotypes that have the interaction with the cell type. **g** Left: UMAP representation showing the distribution of the T-LGLL clonotypes with different numbers of interactions with their non-leukemic counterparts. Right: UMAP representation of the transcriptomes of the selected 18 leukemic T-LGLL clonotypes as seen in Fig. 2b.

variant allele frequency detection sensitivity of 0.5%. It was then sequenced using Illumina HiSeq Reagent Kit v4 100 cycles kit or Illumina MiSeq System using MiSeq 600 cycles kit (Illumina, San Diego, CA, USA).

**Single-cell RNA and TCRαβ-sequencing and data analysis.** Viably frozen cells from 11 T-LGLL samples from 9 T-LGLL patients and 6 age-matched healthy samples were thawed in PBS with 2 mM EDTA and stained with anti CD45+ APC-H7 (Cat#: 560178 BD Biosciences) antibody. CD45+ cells were selected with Sony SH800 (Sony Biotechnology Inc.). Single-cells were partitioned using a Chromium Controller (10X Genomics) and scRNA-seq and TCRαβ-libraries were prepared using Chromium Single Cell 5′ Library & Gel Bead Kit (10X Genomics) (CG000086 Rev D) as done in Kim et al.[56]. Briefly, from individual samples 17,000 cells were suspended in 0.04% BSA and then loaded to a Chromium Single Cell A Chip. After generation of single-cell barcoded cDNA the remaining steps were performed in bulk. To amplify full-length cDNA 14 cycles of PCR (Veriti, Applied Biosystems) were run. Chromium Single Cell Human T cell V(D)J Enrichment Kit (10× Genomics) was used to amplify TCR cDNA. Illumina NovaSeq, S1 flowcell (read length configuration: Read1 = 26, i7 = 8, i5 = 0, Read2 = 91) was used for sequencing gene expression libraries. Illumina HiSeq2500 in Rapid Run (read length configuration: Read1 = 150, i7 = 8, i5 = 0, Read2 = 150) was used for sequencing TCR-enriched libraries. The raw data were processed using Cell Ranger (ver 2.1.1) with GRCh38 as the reference genome. Additional scRNA-seq data from CD45+ sorted samples from patients with chronic myeloid leukemia ($n = 4$), chronic lymphocytic leukemia ($n = 4$), non-small cell lung carcinoma ($n = 1$), and renal cell carcinoma ($n = 3$) were also gathered as stated in Supplementary Data 1.

For the T-LGLL samples, specific quality control thresholds were used for individual samples to retain the T-LGLL samples since T-LGLL samples showed considerable heterogeneity and viability levels during library preparation (Supplementary Data 1). For the healthy samples and the non-leukemic analyses for T-LGLL with comparison data from CLL, CML, RCC, and NSCLC data cells with >15% mitochondrial transcripts, <10% or >50% ribosomal transcripts, <250 or >4,500 expressed genes or <1,000 or >20,000 UMI counts were removed from the analysis. For the non-leukemic analysis, the leukemic cell populations from T-LGLL, CLL, and CML samples were removed as well as a cluster that was specific to T-LGLL and healthy samples produced for this project.

To overcome batch-effect, we used scVI (ver 0.5.0)[57] with default parameters where each sample was treated as a batch. The obtained latent embeddings were then used for graph-based clustering and uniform mainifold approximation and projection (UMAP) dimensionality reduction implemented in Seurat (ver 3.0.0)[58,59]. The datasets were scaled with 3,000 most highly variable genes with the FindVariable-function and ScaleData-functions with default parameters. For each different clustering, the genes related to V(D)J-recombination were removed and the resolution values in FindClusters-function were inspected visually within the range of 0.1–3 with intervals of 0.1, where the chosen values were within 0.2–0.5 to prevent overclustering (for 0.2 for Fig. 1b, 0.5 for Fig. 1c and the same clusters are in Fig. 1e, 0.2 for Fig. 2b, 0.2 for Fig. 3g, 0.5 for Fig. 4a, and 0.3 for Supplementary Fig. 17a). Clusters are named in descending order (cluster 0 contains the most cells) and were annotated by analysis of canonical markers, differentially expressed genes, relationship to other clusters, signature scores, T cell receptor repertoire clonalities, and reference-bases cell-type annotation with SingleR[60](ver 1.2.4) with Blueprint[61] as a reference. For UMAP-dimensionality reductions, the default parameters in RunUMAP-function were used throughout. Pseudotime analyses were done with Slingshot (ver 1.1.4)[36] on unsupervised mode on precalculated UMAP coordinates with default parameters.

Differential expression analyses were performed based on the $t$-test, as suggested by Soneson et al.[62], and $P$-values were adjusted with Bonferroni correction. Enrichment analyses were performed with the up or downregulated genes ($P_{adj} < 0.05$) with hypergeometric testing implemented in ClusterProfiler (3.16.0)[63] with GO- and HALLMARK-categories gathered from MSigDB. GO-

categories were inspected manually, and redundant pathways were removed from visualizations but retained in Supplementary Data 2.

Different scores were calculated with the AddModuleScore-function, as suggested by Tirosh et al.[41], which briefly considers the expression of a given set of genes and subtracts a similarly counted expression of a randomly selected gene set. Cytotoxicity score was calculated with genes defined by Dufva and Pölönen et al.[28], including *GZMA, GZMH, GZMM, PRF1*, and *GNLY*. The IFNγ and NF-κB scores were calculated from genes included the gene sets downloaded from MSigDb HALLMARK-categories (HALLMARK_INTERFERON_GAMMARESPONSE, 200 genes, ver 5.0; and HALLMARK_TNFA_SIGNALING_VIA_NFKB, 200 genes, ver 5.0). HLA II score was calculated with *HLA-DMA, HLA-DMB, HLA-DOA, HLA-DOB, HLA-DPA1, HLA-DPB1, HLA-DQA1, HLA-DQB1, HLA-DQB1-AS1, HLA-DQA2, HLA-DQB2, HLA-DRA, HLA-DRB1*, and *HLA-DRB5*. The scavenging receptor score was calculated with genes included in the Hugo Gene Nomenclature under Scavenger Receptors (SCAR)[64], including *CD14, CD207, CD209, LY75, MRC1, MSR1, MARCO, SCARA3, COLEC12, SCARA5, SCARB1, SCARB2, CD36, CD68, OLR1, CLEC7A, SCARF1, SCARF2, MEGF10, CXCL16, STAB2, STAB1, CD163, CD163L1, AGER, SSC4D, SSC5D*. To visualize gene expressions, scaled expressions were used in the FeaturePlot-function implemented in Seurat, where the order = T option was used.

Ligand–receptor interaction analyses were performed with CellPhoneDB (ver 2.0.0)[43] with default parameters for subsets with at least 50 cells and 1,000 iterations for the permutation testing. The co-stimulatory and coinhibitory receptor–ligand pairs were gathered from Dufva and Pölönen et al.[28].

Somatic variant detection was performed with Vartrix[65] (ver 1.1.0) with default parameters against the whole COSMIC database (ver 86) except indels >10 base pairs in length. Only STAT3 variants associated with CD8+ T-LGLL (Y640F, S614R, N647I, I659L, and D661Y) were retained.

To calculate regulons for T-LGLL clonotypes phenotypes, the SCENIC[27] (ver 1.2.4) vignette was followed with the default parameters.

Heat maps were performed with the ComplexHeatmap package (ver. 2.4.2), where different clustering analyses were performed with Ward's linkage with default parameters and seed as 123. For clustering based on the IFNγ and NF-κB scores, $k$ was chosen as 2 and for the interactome analysis as 4 after visual inspections for values of $k$ between 2–10.

For scTCRαβ-seq, and only TCR productive full-length sequence information were considered and all ambiguous cells with multiple TCRα and/or TCRβ chains were removed. Clones were defined as exact same CDR3 amino acid sequence in both TCRαβ-chains, if available, or just in TCRβ-chain. The clonotypes for individual samples have been named in descending order (clonotype 1 contains the most cells). T-LGLL clonotypes were inferred as stated in the manuscript by manually curating data from scTCRαβ-seq, Vβ flow cytometry, and STAT3 amplicon sequencing data. From scTCRαβ, wild-type T-LGLL clonotype had to explain at least >5% of total TCR repertoire (in any time point, if multiple timepoints present). For patient 1, clonotype 4 was seen in both timepoints in scTCRαβ-seq data but was filtered during quality control in scRNA-seq data in time point 2015.

**Bulk-RNA sequencing and data analysis.** Bulk-RNA-sequencing was performed as described by Savola et al.[32]. Briefly, Qiagen miRNeasy micro kit (cat. no 217084) and SMART-Seq v4 Ultra Low Input RNA Kit (cat. no. 634890) was used to extract RNA. Sequencing was conducted using Illumina Nextera XT kit (FC-131-1096). Data filtering was done using Trimmomatics (filtering parameters leading: 3, trailing: 3, sliding window: 4:15 and minlen: 36). STAR aligner was used for alignment using the human reference genome (Ensembl GRCh38). EdgeR (3.3.3)[66] was used to count the DEGs, where read counts have normalized with the Trimmed Mean of M-values (TMM) method with exact Test-function implemented in edgeR with dispersion = "common" option. Cytotoxicity scores were calculated as geometric means as suggested by Dufva and Pölönen et al[28], with the same genes as

above in scRNA-seq analysis. TCRαβ-sequences were gathered from bulk-RNA-sequencing data with MiXCR (ver 3.0.13)[67] with default parameters.

**TCRβ-sequencing and data analysis**. TCRβ-sequencing from the genomic DNA was conducted with ImmunoSEQ assay by Adaptive Biotechnologies Corp as per manufacturers guidance and as previously described by Savola et al.[32]. Additional TCRβ data from CD8[+] sorted samples from patients with rheumatoid arthritis from diagnosis ($n = 32$), metastatic melanoma from diagnosis ($n = 29$), and healthy control samples ($n = 38$) from peripheral blood and MNC-sorted samples from patients with T-LGLL ($n = 38$) or healthy controls ($n = 785$) from peripheral blood were also gathered as stated in Supplementary Data 1.

Analyses were done with VDJtools (ver 1.2.1)[68], where non-functional clonotypes were removed and diversity indices calculated with CalcDiversityStats-function. To allow reliable diversity metrics, all samples were subsampled to 30,000 reads and samples that had fewer reads were removed from further analyses ($n = 28$; 13 RA samples and 15 T-LGLL samples from Kerr et al.).

TCRs were grouped based on amino acid-level-similarities decided by GLIPH2 (1.0.0)[31], with default parameters and CD8 as reference sets for CD8[+]-sorted samples and CD4CD8 for MNC-sorted samples. To detect antigen-driven clonotypes, the subsampled TCRβ-seq or scTCRαβ-seq samples were inputted individually to GLIPH2. The analysis was repeated also for samples where the non-leukemic repertoire in T-LGLL or to samples without the largest clone for the rest of the cohorts were subsampled to the same read-depth of 30,000 reads to avoid biases. In TCRβ-seq data T-LGLL, the clonotypes in the CD8[+] data explaining >5% of the repertoire, in the MNC data reported in the original publication by Kerr et al.[24], or in scTCRαβ data found as in Fig. 2b were assumed to be T-LGLL clones (Supplementary Data 3). Similarly for other datasets, the antigen drive for the largest clone was analyzed. The presence of antigen drive was defined if GLIPH2 notified a statistically significantly enriched cluster with at least two TCRs against the reference dataset included in GLIPH2.

Epitope predictions were performed with TCRGP (ver 1.0.0)[35] using precalculated models against HLA-A*02 background gathered from the packages GitHub page for each TCRβ identified in the dataset. The tested epitopes were "GILGFVFTL_cdr3b" (from Influenza A M1 $_{GILGFVFTL}$ epitope), "GLCTLVAML_cdr3b" (EBV BMLF1$_{GLCTLVAML}$ epitope), "IPSINVHHY_cdr3b" (CMV pp65$_{IPSINVHHY}$ epitope), "NLVPMVATV_cdr3b" (CMV pp65$_{NLVPMVATV}$ epitope), "RAKFKQLL_cdr3b" (EBV BZLF1$_{RAKFKQLL}$ epitope), "RPRGEVRFL_cdr3b" (HSV2 B7$_{RPRGEVRFL}$ epitope), "TPRVTGGGAM_cdr3b" (CMV pp65$_{TPRVTGGGAM}$ antigen), and "YVLDHLIVV_cdr3b" (EBV BRLF1$_{YVLDHLIVV}$ epitope). The probability needed to be above a cut-off of 0.85 to be considered as specific to the tested epitope.

**HLA genotyping and HLA phenotyping inference from sequencing data**. The healthy samples profiled with scRNA+TCRαβ-seq ($n = 6$) were typed at the Histocompatibility Testing Laboratory, Finnish Red Cross Blood Service accredited by European Federation for Immunogenetics. The HLA specificities were reported based on the current World Health Organization (WHO) nomenclature for the HLA system. The typing for HLA-A, -B, -C, and -DRB1 loci was performed using the Luminex bead array technology together with sequence-specific oligonucleotide probes (Commercial LabType kits RSSO1A, RSSO1B, RSSO1C, RSSO2B1, One Lambda, Los Angeles, CA). The bead array data were interpreted according to the manufacturer's recommendations using the HLA Fusion software 3.2 (One Lambda).

HLA phenotypes were inferred from the paired-end scRNA-seq and bulk-RNA-seq data with PHLAT (v 1.1) and bowtie (v 2.7.0) with default parameters, ran on paired-end mode. Convincingly, PHLAT arrived at the same six-digit allele as in HLA-A, -B, -C and -DRB1 loci in 47/48 (97.91%) of the alleles in the healthy donors, where the only difference was in one individual where the HLA-C*07:02 was predicted to be HLA-C*4:01. In the T-LGLL samples profiled with scRNA-seq, we had two time series samples to consider how reproducible the algorithm is for different samples from the same individual. When the two-digit accuracy was considered, the agreement between different timepoints for Pt1 was 8/8 (100%) and for Pt2 7/8 (87.5%), where the different HLAs were HLA-B*07 and HLA-B*40. When considering six-digit accuracy, the accuracy was 7/8 (87.5%) for Pt1 and 5/8 (62.5%) for Pt2.

The full HLA-genotype and HLA inferred phenotypes can be found in Supplementary Data 1 and 3.

**Plasma cytokine analysis**. A multiplexed Proseek Multiplex Inflammation I (Olink Biosciences) panel including 92 proteins from 17 plasma samples from 9 T-LGLL patients and 8 healthy donors were gathered from Savola et al.[42] publication. The differentially expressed cytokines were calculated from the normalized protein expression units (NPX) with Mann-Whitney test and corrected with Benjamini-Hochberg.

**Functional validations**. Three different experiments were done for the functional validation cohort: cytotoxicity and cytokine secretion, proliferation, and phagocytosis assays. On day 1, viably frozen cells were thawed and plated on 96 well plates and cultured overnight in complete RPMI. For the Cytotoxicity and cytokine secretion assay, 20 ul of CD3 APC UCHT1 (BD Cat. 555335), 1.8 ul α-CD49d

(1:10 dilution, BD, Cat. 340976), 1.8 ul of α-CD28 (1:10 dilution, BD, Cat. 340975), 10 ul of CD107a FITC (BD, Cat. 555800), 10 ul of CD107b FITC (BD, Cat. 555804), and 0.36 ul of GolgiStop (BD, Cat. 554724) were added to a total volume of 150 ul RPMI the next day and cells were incubated overnight at 37 °C. The following day cells were washed with 1 ml of PBS-EDTA-BSA and stained with 20 ul of CD3 UCHT1 APC, 0.5 ul of CD3 SK7 APC (BD, Cat. 345767), 2.5 ul of CD57 PE (BD Cat. 560844), 0.8 ul of CD8 PE-CY7 (BD, Cat. 335822), 1 ul of CD45 V500 (BD, Cat. 655873), 3 ul of TNF V450 (BD, Cat. 561311), 3 ul of IFNG V450 (BD, Cat. 560371), 0.5 ul of Granzyme A AlexaFluor700 (Biolegend Cat. 507210), 0.9 ul of Granzyme B AlexaFluor700 (BD, Cat. 561016, 560213), 2.5 ul of Perforin 1 PerCP-Cy5.5 (BD, Cat. 563762). The staining antibodies were not diluted and used according to the manufacturer's guidelines.

For the proliferation assay, cells were washed and incubated with CFSE Cell Division tracker kit (Biolegend Cat. 423801) for 20 min at 37 °C protected from light. Fluorescence was quenched by adding RPMI, washed, resuspended in complete RPMI and incubated at room temperature for 10 min. Cells were then added to wells, pre-coated with CD3 okt-3 (BD, Cat. 555329) on the day before. Stimulants for cell proliferation were added as follows: α-CD49d + α-CD28 or LPS (Sigma Aldrich cat. - L2018) or LPS + R848 (Resiquimod, Sigma Aldrich Cat. SML0196) and incubated for 72 h at 37 °C. On Day 4, cells were washed with 1 ml of PBS-EDTA-BSA and stained with 2.5 ul of CD57 PE (BD Cat. 560844), 0.8 ul of CD8 (PE-Cy7), 1 ul of CD45 (V500), 5 ul of CD14 (APC-Cy7), 2.5 ul of CD16 (PerCP-Cy5.5), 2.5 ul of CD19 (V450), and 1 ul of CD3 (APC).

For the phagocytosis assay, on day 2 cells were incubated with FluoSpheres fluorescent beads (FluoSpheres Carboxylate-Modified Microspheres, 1.0 μm, yellow-green fluorescent (505/515), 2% solids—F8823, Thermofisher) at a concentration of cells to beads ratio of 1:10. Cells were incubated for 30 min at 37 °C protected from light in only a serum-containing medium. Cells were trypsinized and washed with 1 ml of PBS-EDTA-BSA and stained with the following markers—2.5 ul of CD14 Pe-Cy7 (BD Cat. 562698), 2.5 ul of CD16 PerCP-Cy5.5 (BD Cat. 560717), and 2 ul of CD45 APC-H7 (BD Cat. 641417). All stained samples were analyzed with Cells were acquired on BD FACSVerse and FlowJo software (Version v10.7, Becton Dickinson).

**Statistical testing**. P-values were calculated with nonparametric tests, including Mann-Whitney test (two groups), Kruskal-Wallis test (more than two groups), and Fisher's exact test where the alternative hypotheses are reported. P-values were corrected with Benjamini-Hochberg adjustment. All calculations were done with R (4.0.2) or Python (3.7.4).

**Data visualization**. In the box plots, center line corresponds to the median, the box corresponds to the interquartile range (IQR), and whiskers 1.5 × IQR, while outlier points are plotted individually where present.

**Reporting summary**. Further information on research design is available in the Nature Research Reporting Summary linked to this article.

## Data availability

The processed scRNA-sequencing data for both the T-LGLL and healthy samples generated in this study are available at ArrayExpression under accession code E-MTAB-11170. The raw scRNA-sequencing and bulk-RNA-sequencing are available in the European Genome-Phenome Archive under accession code EGAS00001005297. The TCRαβ-sequencing data, TCRβ-sequencing data, and Seurat-objects are available at Zenodo under: https://doi.org/10.5281/zenodo.4739231 [https://zenodo.org/record/4739231] with restricted access due to GDPR regulations and data can be accessed by placing a request via Zenodo. The publicly available scRNA+TCRαβ-sequencing and TCRβ-sequencing data used in this study are listed in Supplementary Data 1. Source data are provided with this manuscript. Source data are provided with this paper.

## Code availability

The code to reproduce the key findings is available in Github [https://github.com/janihuuh/cd8_tlgll_manu] (v1, https://zenodo.org/badge/latestdoi/356225989 [https://zenodo.org/record/5715103#.YaS9p_FBzGw]).

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

## Acknowledgements

This project was funded by the European Research Council (Project: M-IMM 647355), Academy of Finland Heal-Art consortium (314442), and ERA PerMed (JAKSTAT-TARGET consortium), Sigrid Juselius Foundation, Signe and Ane Gyllenberg Foundation, Helsinki Institute for Life Science, Cancer Foundation Finland, and Ministry of Education, Culture, Sports, Science and Technology of Japan (Kaken 20K08709). J.H. was supported by Finnish Hematology Association, Blood Disease Research Foundation, Helsinki Institute for Life Science, Biomedicum Helsinki Foundation, Finnish Medical Foundation. T.L. was supported by Academy of Finland (Decision 311081). RZ was supported by Italian Association for Cancer Research (AIRC #20216). F.I. was supported by KAKEN 20K08709. from Grant-in-Aid for Scientific Research from the Ministry of Education, Culture, Sports, Science and Technology of Japan. Single-cell RNA sequencing and amplicon sequencing were performed at the Institute for Molecular Medicine Finland FIMM Technology Centre and Finnish Functional Genomics Centre, Turku Bioscience, which are supported by Biocenter Finland. We acknowledge Jay Klievink, Sofie Lundgren, Anita Kumari, and Ella Piekkari for the processing of samples from healthy controls. We acknowledge Emmi Rehn for providing a modified script of GLIPH that was used for the previous version of the findings. We acknowledge the computational resources provided by the Aalto Science-IT project.

## Author contributions

J.H., D.B., T.Ke., and S.M. conceived the study. C.K., H.R., R.Z., M.H., T.Kas., C.G., F.I., T.Lo., T.Ka., and J.M. recruited the patients or handled patient samples, and D.B., T.L., C.K., T.Kas., T.B., and A.T. profiled the samples. J.H. and D.B. designed and performed the data analysis and acquired the publicly available comparison data with help from M.K., J.T., C.G., and H.L. D.B., M.S., T.Ke., and S.M. designed and D.B. and T.Kas. performed the functional validations. J.H. and D.B. drafted the paper and designed the figures with T.Ke. and S.M. with contributions from all authors. T.Ke. and S.M. jointly supervised the project.

## Competing interests

T.L. is a member of the scientific advisory boards and hold stock options of Keystone Nano, Dren Bio, and Kymera Therapeutics (not related to this project). S.M. has received honoraria and research funding from BMS and research funding from Novartis and Pfizer (not related to this project). The remaining authors declare no competing interests.
