## [Peer Review File · Nature Communications]

Single-cell transcriptomics identifies synergistic role of leukemic and non-leukemic immune repertoires in CD8+ T cell Large Granular Lymphocytic LeukemiaREVIEWER COMMENTS

Reviewer #1 (Remarks to the Author):

The authors conduct a thorough and informative analysis on CD8+ T-LGL using single-cell transcriptomic methodology, providing important insight into the biology of the disease. The findings provide a basis for a potential paradigm shift in how patients with T-LGL might be treated. Notably, they align with and explain empiric clinical observations long associated with this disease. We find the methodology sound, and the conclusions drawn by the authors are well-founded.

Reviewer #2 (Remarks to the Author):

The authors present a comprehensive study in which they seek to characterize both the leukemic and non-leukemic cells in the context of T-cell Large Granular Lymphocytic Leukemia (T-LGLL) via single cell paired RNA expression & TCR-alpha/beta profiling.

The authors show that within a given individual there exists TCR homology between the TCRs of T-LGLL clonotypes and non-leukemia cells; in addition, they show that the non-leukemic TCRs were more mature and restricted as compared to healthy controls. These findings are well presented in the manuscript and are consistent with the notion of an antigen driven response in most of the T-LGLL cases reviewed. Additionally, the authors identified interesting differences between STAT3 mutated vs wild-type T-LGLL, particularly around the concepts of T-cell activation, cytotoxicity, and exhaustion. Further, the authors explored putative epitopes for the TCRs of T-LGLL clonotypes using a variety of machine learning and laboratory methods.

This reviewer's areas of expertise lie in the analytics of single cell data of various forms (RNA expression, TCR, and others) along with associated correlative analyses. From that perspective, I find this manuscript to be very well done and comprehensive. I have no specific critiques or comments that I would ask the authors to address. Although I do think that the work significantly expands our understanding of T-LGLL in the context of the larger scale of the immune system, I defer disease specific questions to reviewers with specific expertise in hematologic malignancies.

Reviewer #3 (Remarks to the Author):

In this study, the authors compare the transcriptional programs of leukemic and non-leukemic T cells from patients with large granular lymphocytic leukemia. This is an interesting and comprehensive study, with a dataset that will surely be of use to the field. Despite this, I found it challenging to weed out the main, important findings from the excess of modestly interesting, superficial data that did not directly contribute to the impact of the study. There are some interesting biological findings, and the study could benefit from focusing more on these solid findings while minimizing distracting data elements. Major and minor critiques are as follows:

Major

1. Since it seems leukemic T cells have lost their ability to signal through the TCR, does the downstream effector programming really matter?
2. Better figure labeling and more information in the legends would help the reader to better interpret the figures.
3. In the analysis of other immune cells in T-LGLL vs. other patients, isn't it possible that these cells are just responding to a tumor? T-LGLL is a blood cancer, meaning the anti-tumor immune response is less "sequestered" than in solid malignancies. It seems that to truly understand the impact of T-LGLL on the immune compartment, the NSCLC and RCC patients should either be removed or separated into their own "cohort" in order to compare what is just a byproduct of a blood cancer (using the CLL and CML patients) vs. truly interesting biology.

4. The entire antigen-driven analysis needs to be tightened up to focus on only the key information that is new or meaningful for the story, and proper "control" analyses need to be performed to decipher meaningful biology from artifact.

a. Regarding the analyses of "antigen-drive", this could be influenced by the non-leukemic library size. An analysis should be performed to show that a lower non-leukemic library size is not associated with a lower "antigen-drive"

b. Lines 242-249 discuss the likely shared antigen specificity between leukemic and non-leukemic T cells within the same individual. It is unclear what is the novelty of this finding, as it is known and expected that polyclonal antigen-specific T cell responses exist.

c. "Interestingly, the antigen-driven clonotypes were more frequently observed in the healthy controls' TCRb repertoires than the non-antigen-driven clonotypes, suggesting that antigen-driven clonotypes could recognize commonly encountered antigens." -This is both known and expected and does not add to the message of the study

d. The analysis using TCRGP is going to be severely biased toward common HLA alleles that present common viral antigens. Without deconvoluting the impact of HLA heterogeneity on the TCRGP results within this cohort, the false negative rate is immeasurable and the results are uninterpretable.

e. "...the dominant mutated STAT3 clone was replaced by an antigen-driven wild-type STAT3 clone..." -Is this a leukemic clone with the same TCRa/b? If so, this seems like a very important and interesting finding that should be highlighted more.

Minor

1. "Next we analyzed the presence of the 200 T-LGLL clonotypes in a cohort of 785 healthy donors' TCRb repertoires. Less than half (36.6%) of the clonotypes were found in the healthy TCR repertoires, and they explained <1% of the healthy repertoire." -Is this meaningful, given the extreme TCR repertoire heterogeneity between humans who are not HLA-matched?

2. "Non-leukemic T cell populations are more mature and clonal than in other cancers, RA, and healthy controls." -Couldn't this just reflect anti-tumor responses to a blood cancer that is readily seen in lymph nodes and the periphery?

3. Line 331: "...supporting the distinction between the pathogenesis of T-LGLL with mutated and wild-type STAT3." -The data presented here do not show any difference in pathogenesis as a function of STAT3 mutation.

Reviewer #1 (Remarks to the Author):

The authors conduct a thorough and informative analysis on CD8+ T-LGL using single-cell transcriptomic methodology, providing important insight into the biology of the disease. The findings provide a basis for a potential paradigm shift in how patients with T-LGL might be treated. Notably, they align with and explain empiric clinical observations long associated with this disease. We find the methodology sound, and the conclusions drawn by the authors are well-founded.

Our response: We thank the reviewer for the careful evaluation of our work and the kind feedback.

Reviewer #2 (Remarks to the Author):

The authors present a comprehensive study in which they seek to characterize both the leukemic and non-leukemic cells in the context of T-cell Large Granular Lymphocytic Leukemia (T-LGLL) via single cell paired RNA expression & TCR-alpha/beta profiling.

The authors show that within a given individual there exists TCR homology between the TCRs of T-LGLL clonotypes and non-leukemia cells; in addition, they show that the non-leukemic TCRs were more mature and restricted as compared to healthy controls. These findings are well presented in the manuscript and are consistent with the notion of an antigen driven response in most of the T-LGLL cases reviewed. Additionally, the authors identified interesting differences between STAT3 mutated vs wild-type T-LGLL, particularly around the concepts of T-cell activation, cytotoxicity, and exhaustion. Further, the authors explored putative epitopes for the TCRs of T-LGLL clonotypes using a variety of machine learning and laboratory methods.

We thank the reviewer for a succinct recap of our work.

This reviewer's areas of expertise lie in the analytics of single cell data of various forms (RNA expression, TCR, and others) along with associated correlative analyses. From that perspective, I find this manuscript to be very well done and comprehensive. I have no specific critiques or comments that I would ask the authors to address. Although I do think that the work significantly expands our understanding of T-LGLL in the context of the larger scale of the immune system, I defer disease-specific questions to reviewers with specific expertise in hematologic malignancies.

Our response: We thank the reviewer for the careful evaluation of our work and the positive comments on the methods and findings.

Reviewer #3 (Remarks to the Author):

In this study, the authors compare the transcriptional programs of leukemic and non-leukemic T cells from patients with large granular lymphocytic leukemia. This is an interesting and comprehensive study, with a dataset that will surely be of use to the field.

We thank the reviewer for the positive comments.

Despite this, I found it challenging to weed out the main, important findings from the excess of modestly interesting, superficial data that did not directly contribute to the impact of the study. There are some interesting biological findings, and the study could benefit from focusing more on these solid findings while minimizing distracting data elements. Major and minor critiques are as follows:

We have improved our manuscript by removing unnecessary sentences and have moved some panels from the main **Figures** to the **Supplementary Figures** and have answered the major and minor critiques point by point below.

Major

1. Since it seems leukemic T cells have lost their ability to signal through the TCR, does the downstream effector programming really matter?

Our response: We thank the reviewer for raising this important comment. We would like to clarify that in our experiments, we did not have any direct antigenic TCR stimulation, but rather unspecific TCR and co-stimulation via CD3, CD28, and CD49, and therefore we have deferred from commenting on whether T cells have lost their ability to signal through TCR in the manuscript.

In our hypothesis of T-LGLL (and seen e.g., in the review by Lamy et al¹), we think that T-LGLL is caused by an aberrant oligoclonal immune response against a patient-specific antigen, which primarily causes a strong signaling cascade through the TCR, which could lead to downstream activation of JAK-STAT signaling, as seen in both patients with mutated and wild type *STAT3*². The upregulated JAK-STAT signaling is then important in sustaining the proliferation and anti-apoptotic properties of T-LGLL cells³. We believe that if the T-LGLL cells have indeed lost their ability to signal through the TCR, this event is secondary to the primary event. However, T-LGLL is thought to develop over a long period, and unfortunately, no time-series data before the initiation of T-LGLL is available.

Nevertheless, the upregulation of JAK/STAT is still seen throughout in different time points, and its activation strength may be associated with therapy responses^{4,5}. Also, CD52, which impairs the phosphorylation of TCR-associated LCK and ZAP70⁶, is highly expressed in T-LGLL as shown in previous publications⁷ and in our samples and has offered some therapeutic potential in a phase II trial in T-LGLL⁸. Therefore, we would argue that by studying the downstream effector programming we could learn mechanistic insights on how to attenuate the T-LGLL process with different regimens.

2. Better figure labeling and more information in the legends would help the reader to better interpret the figures.

Our response: We thank the reviewer for this comment. We have added more detailed information in the legends throughout to aid the reader in the interpretation of the main **Figures** (specifically **Figures 1a-i, 2a-k, 3a-f, 4a-d, 5a-e, and 6a-g**) and **Supplementary Figures** (specifically **Supplementary Figures 5-11 and 13-15**). For example, different clusters and axes are better explained in the figures and each legend includes the number of samples used in the experiment.

3. In the analysis of other immune cells in T-LGLL vs. other patients, isn't it possible that these cells are just responding to a tumor? T-LGLL is a blood cancer, meaning the anti-tumor immune response is less "sequestered" than in solid malignancies. It seems that to truly understand the impact of T-LGLL on the immune compartment, the NSCLC and RCC patients should either be removed or separated into their own "cohort" in order to compare what is just a byproduct of a blood cancer (using the CLL and CML patients) vs. truly interesting biology.

Our response: We thank the reviewer for pointing out the role of anti-tumor immunity in T-LGLL. T-LGLL differs from both other hematologic and solid cancers, and it can also be considered as "benign" autoimmune disease albeit of somatic mutations and large T cell clones (recently reviewed by us⁹, Mustjoki and Young, *New Engl J Med* 2021). Unlike in other hematologic malignancies, there is little evidence of anti-tumor response against the T-LGLL itself, and T-LGLL can arise secondary to a malignancy¹. Therefore, we included in our analysis samples from healthy controls, patients with an autoimmune disease (RA) and patients with cancer to compare the T-cell responses in all these entities.

However, we do agree with the reviewer that the anti-tumor responses in hematologic and solid malignancies can be considerably different. Therefore, we have re-analyzed the data by removing the NSCLC ($n=2$) and RCC ($n=1$) samples from the analysis seen in **Fig 4b, Supplementary Fig.14, Supplementary Fig. 15b-c, and Supplementary Fig. 16c**. This resulted in a smaller study population (before $n=20$, now $n=17$, reduction of 15%), which could also hamper the interpretation. Nevertheless, we conclude that the main biological signal remained similar.

More specifically, in the analysis based on cell abundances in non-leukemic repertoire in T-LGLL in comparison to other blood cancers, we found that the CD4+ T-cell effector memory clusters (clusters 2 and 15) are expanded similarly as in the original comparison to all cancers (revised **Fig. 4b** includes now both comparisons). Further, the suppressed clusters include a small CD8+ T-cell effector cluster (cluster 14) and a cytotoxic, CD56dim NK-cell cluster (cluster 3), possibly reflecting the anti-tumor immune responses in other hematologic cancers (**Review Figure 1a-b**). We have added the **Review Figure 1a** as a panel in the **Fig. 4b** and the exact P -values of the population abundancies to **Supplementary Table 2** and modified the text accordingly (lines 348-350).

Review Figure 1: Differentially abundant non-leukemic immune subtypes

Differentially abundant non-leukemic clusters (from **Fig. 4a**) between patients with **a**) T-LGLL ($n=9$) and patients with other cancers ($n=11$ [CLL $n=4$, CML $n=4$, NSCLC $n=1$, RCC $n=2$]), and **b**) T-LGLL and patients with blood cancers ($n=8$ [CLL $n=4$, CML $n=4$]). The horizontal line indicates $P=0.05$, as calculated with a Mann-Whitney test.

The figures can be seen in **Fig. 4b** and **Supplementary Fig. 13d**.

In the analysis of differentially expressed genes (DEGs) of the non-leukemic repertoire in T-LGLL in comparison to other blood cancers (as in **Supplementary Figs. 14a-b, 15b-c, and 16c**), we found that removing the solid cancer samples had some changes for the DEGs, although the concordance between these two analyses was very good as measured by the Spearman's correlation coefficient of the fold-change between the two comparisons (T-LGLL vs all cancers and T-LGLL vs blood cancers, **Review Figure 2a**). Especially, one of the main findings (upregulated IFN γ response, as seen by e.g., upregulation of HLA-machinery [*B2M*, *TAP1*], interferon associated genes [*IFITM2/3*, *ISG2*], proteasome activation [*PSME1/2*, *PSMB9/10*], and cytotoxicity genes [*GZMA*] in the non-leukemic repertoire of T-LGLL) was even more pronounced in some immune subpopulations when the comparison was done only on patients with different blood cancers (**Review Figure 2b-c**). We have added the **Review Figure 2c** as a **Supplementary Fig. 14b** and the exact P -values of the DEGs to **Supplementary Table 2** and modified the text accordingly (line 380).

Review Figure 2: Differentially expressed genes between non-leukemic immune subtypes

a) Correlation plot between the average fold-change (FC) between the differentially expressed genes between the two different comparisons (T-LGLL vs solid cancers and T-LGLL vs blood cancers). The correlation coefficients and P -values were calculated with Spearman's rank correlation. **b-c)** Expression of selected differentially expressed genes ($P_{adj} < 0.05$, Bonferroni corrected t-test) between non-leukemic immune cells from **b)** patients with T-LGLL ($n=9$) and patient with other cancers ($n=11$ [CLL $n=4$, CML $n=4$, NSCLC $n=1$, RCC $n=2$]) and **c)** patients with T-LGLL ($n=9$) and patient with other blood cancers ($n=8$ [CLL $n=4$, CML $n=4$]). Values are presented as log2 fold-change (log2fc).

The figures can be seen in **Supplementary Fig. 14a-b**.

Also, the upregulated pathways comparing T-LGLL only to blood cancers were retained. The most upregulated pathways in non-leukemic cells in T-LGLL included IFN γ -response (upregulated in 12/16 subsets vs. other cancers, 9/15 vs other blood

cancers), IFN α -response (9/16 vs. other cancers, 8/15 vs other blood cancers) (Review Figure 3a-b). We have added the Review Figure 3b as a Supplementary Fig. 14d and the exact P -values of the DEGs and the pathways to Supplementary Table 2 and have modified the text accordingly (lines 391-394).

Review Figure 3: Differentially expressed genes between non-leukemic immune subtypes

a-b) Upregulated HALLMARK-category pathways ($P_{adj}<0.05$, hypergeometric test on differentially expressed genes) in non-leukemic cells from **a)** patients with T-LGLL ($n=9$) and patient with other cancers ($n=11$) and **b)** patients with T-LGLL ($n=9$) and patient with other blood cancers ($n=8$).

The figures can be seen in **Supplementary Fig. 14c-d**.

Regarding the analysis on the monocytes (as in **Supplementary Fig. 15b-c**), the analysis was done on T-LGLL vs other conditions (T-LGLL vs CLL+CML+NSCLC+RCC+healthy) to tease out the most unique signature in monocytes in T-LGLL. We have now also performed a comparison of T-LGLL vs other cancers and T-LGLL vs blood cancers. As seen in the **Review Figure 2a**, the correlation between the DEGs between monocytes in these comparisons was statistically significant, but their correlation coefficients were not among the best (varied between 0.4 and 0.51).

When analyzing only T-LGLL against the patients with other cancers (CLL+CML+RCC+NSCLC), the ranking based on the number of DEGs resulted in 2/3 of the monocyte populations to be found in the top 5 most altered ones and 1/3 when comparing T-LGLL against patients with other blood cancers (CLL+CML), instead of the 3/3 in the analysis including all of the conditions (CLL+CML+RCC+NSCLC+healthy) (**Review Figure 4a**). Nevertheless, the most upregulated genes in T-LGLL remained the same, including different class I and class II HLAs (**Review Figure 4b-c**), which could be due to increased IFN γ response seen in the samples, thus keeping with the concordance of our analyses. We have added the **Review Figure 4a-c** as **Supplementary Fig. 15d-e** and the exact P -values of the DEGs to **Supplementary Table 2**.

Review Figure 4: Differentially expressed genes between monocytes

a) Number of differentially expressed genes (DEGs, $P_{adj} < 0.05$, Bonferroni corrected t-test) in each cell cluster between patients with T-LGLL and patients with other cancers (CLL $n=4$, CML $n=4$, NSCLC $n=1$, RCC $n=2$, **left**) and patients with T-LGLL and patients with other blood cancers ($n=8$, **right**) **b-c**) Differentially expressed genes ($P_{adj} < 0.05$, Bonferroni corrected t-test) between different monocyte clusters between **b**) patients with T-LGLL and patients with other cancers and **c**) patients with T-LGLL and patients with other blood cancers. The top 20 genes are labeled.

The figures can be seen in **Supplementary Fig. 15b-e**

Also, the scavenger receptor score in different monocyte populations (as in **Supplementary Fig. 16c**) was higher in T-LGLL when compared to healthy donors in all monocyte comparisons. In all comparisons, T-LGLL had a lower scavenger score than patients with NSCLC ($n=1$), but higher scores than patients with CLL in 3/3 of monocyte clusters and higher than CML in 2/3 clusters. (**Review Figure 5a**). We have added the **Review Figure 5a** as a **Supplementary Fig. 16c** and the exact P -values of the DEGs to **Supplementary Table 2** and the results accordingly (line 437).

a

Review Figure 5: The scavenger receptor score in monocytes

a) Scavenger receptor score of different monocyte clusters in T-LGLL and in comparison to other conditions (CLL $n=4$, CML $n=4$, NSCLC $n=1$, RCC $n=2$, healthy $n=6$). P -values were calculated with a Kruskal-Wallis test.

The figure can be seen in **Supplementary Fig. 16c**.

In the **Fig. 4a**, **Fig. 5d**, **Fig. 6c**, **Supplementary Fig. 13a**, and **Supplementary Fig. 14e** the samples from patients with solid cancers are already visible as their own cohort as the reviewer suggested.

4. The entire antigen-driven analysis needs to be tightened up to focus on only the key information that is new or meaningful for the story, and proper “control” analyses need to be performed to decipher meaningful biology from artifact.

Our response: We thank the reviewer for the comment which led us to improve the main text by, e.g., starting each paragraph with reasoning why each analysis was done and added concluding statements at the end of paragraphs and to the whole section. We have also edited the main **Figures** by moving panels with more detailed data to the **Supplementary Figures**.

We have also provided point-by-point answers and additional control analyses, including e.g., inference of HLA-phenotypes and normalizing the non-leukemic library sizes, to decipher the meaningful biology from artifact.

a. Regarding the analyses of “antigen-drive”, this could be influenced by the non-leukemic library size. An analysis should be performed to show that a lower non-leukemic library size is not associated with a lower “antigen-drive”

Our response: We agree with the reviewer's comment. We have already performed the analysis in a way which is indifferent to the library size, as we have subsampled the samples to the same read depth, 30,000 reads per sample, to avoid bias. This threshold of 30,000 was chosen based on the read depth of our samples, which were done with the “Survey” level with the Adaptive Biotech technologies, which resulted in a minimum of 30,000 reads in a sample.

Based on the reviewers' comment, we also subsampled the non-leukemic repertoires in the T-LGLL cases, healthy donors, and patients with other disorders to 30,000 reads when available. The overall findings remained the same, and the T-

LGLL cases had the largest proportion of antigen-driven samples, and this was statistically significant when compared to other cohorts ($P < 0.05$, Fisher's exact one-sided test, **Review Figure 6a-b**).

Review Figure 6: The antigen drive in non-leukemic library size normalized samples

a-b) Presence of antigen-drive (i.e., whether the largest clonotypes have shared amino acid-level similarities with the rest of the TCR repertoire) in T-LGLL (mononuclear cell [MNC]-sorted $n=17$, CD8+-sorted $n=8$), metastatic melanoma sampled from blood (SKCM, $n=29$), rheumatoid arthritis (RA, $n=32$), and healthy controls (HC, MNC-sorted $n=785$, CD8+-sorted $n=38$). T-LGLL patients had more antigen-driven cases than the rest of the conditions ($P < 0.05$, Fisher's one-sided exact test). In **a**), all the reads from samples were subsampled to the same read-depth (30,000), as in **b**) only the reads from non-leukemic clones (or by removal of the largest clone in the non-T-LGLL cases) were downsampled to the same read-depth (30,000 reads per sample).

The figure can be seen in **Fig. 3b** and **Supplementary Fig. 10a**

We have added the **Review Figure 6a** as a **Supplementary Fig. 10a** and the results to **Supplementary Table 3** and have stated this more clearly in the in the **Results** (lines 256-261, 266-270) and the **Methods** section (lines 719-724).

b. Lines 242-249 discuss the likely shared antigen specificity between leukemic and non-leukemic T cells within the same individual. It is unclear what is the novelty of this finding, as it is known and expected that polyclonal antigen-specific T cell responses exist.

Our response: We thank the reviewer for raising this issue and would like to clarify why we think this is a novel finding. While we agree with the reviewer that polyclonal and oligoclonal T-cell responses exist in the context of immune response in healthy, we would like to underpin that it is currently unknown whether the response in T-LGLL is oligo- or polyclonal.

We think that our results from the antigen drive support the view that the polyclonal response against an antigen is the initiating event, and the *STAT3* mutations are probably later events, solidifying the clonal dominance. Somewhat similar findings have been suggested in a previous publication as well¹⁰, but our results provide a more sophisticated large-scale analysis utilizing similarity analysis of the TCR-repertoire.

We have rephrased the **Discussion** accordingly (lines 516-520).

c. *“Interestingly, the antigen-driven clonotypes were more frequently observed in the healthy controls’ TCRb repertoires than the non-antigen-driven clonotypes, suggesting that antigen-driven clonotypes could recognize commonly encountered antigens.”*

-This is both known and expected and does not add to the message of the study

Our response: We thank the reviewer for raising this issue and would like to clarify why we think finding antigen-driven clonotypes more frequently in the healthy is of interest and a novel finding regarding T-LGLL.

Although we agree that it is known that T-LGLL TCRs can be found in healthy individuals as well^{10,11}, it has been noted to appear rarely. Hence, we consider it is interesting that antigen driven clonotypes are more preferably found in the healthy, even given the HLA-mismatch discussed later by the reviewer.

We did not find the results expected, as although LGL expansions have been associated with common viral infections - such as CMV and EBV especially in post-transplant settings^{12,13} – such associations can be made rarely in non-transplanted patients. Therefore, we think that our finding of the polyclonal immune response in T-LGLL against a possibly abundantly found epitope is of interest to better understand the disease.

We have rephrased the **Discussion** to better explain the importance and novelty of the findings (lines 516-520).

d. The analysis using TCRGP is going to be severely biased toward common HLA alleles that present common viral antigens. Without deconvoluting the impact of HLA heterogeneity on the TCRGP results within this cohort, the false negative rate is immeasurable and the results are uninterpretable.

Our response: The models used in TCRGP were from HLA-A*02 backgrounds, which is the most prevalent in the Caucasian population, and hence we argue that our results are more biased towards overestimating than underestimating the presence of clones targeting viral epitopes. Overall, the key finding was that the most of the T-LGLL clonotypes do not target these known epitopes. As there are limited number of TCR-pMHC pairings where the peptide is the same, but the presenting HLA is different¹⁴, we cannot unfortunately make estimations what are the false-positive and false-negative rates with the used TCRGP models.

Nevertheless, from our scRNAseq and bulk-RNAseq data, we were able to determine the HLAs with a specific algorithm, PHLAT¹⁵. As it has not been published in a setting of scRNA-seq, we tried to validate its usefulness with two experiments.

- i) We have HLA-genotype information provided by the Finnish Blood Service from the 6 healthy controls ($n=6$) from HLA-A, HLA-B, HLA-C, and HLA-DRB1, which we used as a reference to understand the performance of the HLA-inference tool. Convincingly, PHLAT arrived at the same six-digit allele as in HLA-A, -B, -C, and -DRB1 loci in 47/48 (97.91%) of the alleles in the healthy donors, where the only difference was in the HLA-C locus in one individual where the HLA-C 07:02 was predicted to be HLA-C 4:01 (**Review Table 1**).
- ii) In the T-LGLL samples profiled with scRNA-seq, we had two-time series samples to consider how reproducible the algorithm is for different samples from the same individual. When the two-digit accuracy was considered, the agreement between different time points for Pt1 was 8/8 (100%) and for 2 Pt 7/8 (87.5%), where the different HLAs were HLA-B 07 and HLA-B 40. When considering six-digit accuracy, the accuracy was 7/8 (87.5%) for Pt1 and 5/8 (62.5%) for Pt2 (**Review Table 2**).

Patient	timepoint	HLA-A inferred	HLA-A genotyped	HLA-B inferred	HLA-B genotyped	HLA-C inferred	HLA-C genotyped	HLA-DRB1 inferred	HLA-DRB1 genotyped	patient group	type
HC1	1	2:01	2:01	35:01:00	35:01:00	4:01	4:01	1:01	1:01	Healthy	scRNA-seq
HC1	1	11:01	11:01	44:02:00	44:02:00	5:01	5:01	4:04	4:04	Healthy	scRNA-seq
HC2	1	3:01	3:01	7:02	7:02	4:01	4:01	1:01	1:01	Healthy	scRNA-seq
HC2	1	3:01	-	35:01:00	35:01:00	4:01	7:02	15:01	15:01	Healthy	scRNA-seq
HC3	1	2:01	2:01	27:05:00	27:05:00	2:02	2:02	1:01	1:01	Healthy	scRNA-seq
HC3	1	2:01	-	27:05:00	-	2:02	-	8:01	8:01	Healthy	scRNA-seq
HC4	1	2:01	2:01	8:01	8:01	7:01	7:01	3:01	3:01	Healthy	scRNA-seq
HC4	1	23:01	23:01	49:01:00	49:01:00	7:01	-	13:01	13:01	Healthy	scRNA-seq
HC5	1	2:01	2:01	15:01	15:01	3:03	3:03	7:01	7:01	Healthy	scRNA-seq
HC5	1	68:01:00	68:01:00	35:01:00	35:01:00	3:03	-	13:01	13:01	Healthy	scRNA-seq
HC6	1	2:01	2:01	27:05:00	27:05:00	1:02	1:02	4:03	4:03	Healthy	scRNA-seq
HC6	1	11:01	11:01	44:02:00	44:02:00	2:02	2:02	12:01	12:01	Healthy	scRNA-seq

Review Table 1: The analyzed HLA genotypes and inferred HLA phenotypes

The agreement between the genotyped and inferred HLA phenotypes with PHLAT. Red values indicate the difference between two-digit accuracies between the two different methods.

Patient	timepoint	HLA-A inferred	HLA-A genotyped	HLA-B inferred	HLA-B genotyped	HLA-C inferred	HLA-C genotyped	HLA-DRB1 inferred	HLA-DRB1 genotyped	patient group	type
Pt1	1	1:01:01		37:01:01		3:04:01		10:01:03		T-LGLL	scRNA-seq
Pt1	1	24:02:40		40:01:08		6:02:01		13:02:01		T-LGLL	scRNA-seq
Pt1	2	1:01:26		7:02:03		3:04:01		10:01:01		T-LGLL	scRNA-seq
Pt1	2	24:61		37:01:01		6:02:01		13:02:01		T-LGLL	scRNA-seq
Pt2	1	2:01:01		40:11:02		1:02		04:66		T-LGLL	scRNA-seq
Pt2	1	32:01:01		56:01:01		3:04		9:01:02		T-LGLL	scRNA-seq
Pt2	2	2:01:01		40:01:02		1:02		4:03:01		T-LGLL	scRNA-seq
Pt2	2	32:01:01		56:01:01		3:04		9:01:02		T-LGLL	scRNA-seq

Review Table 2: The inferred HLA phenotypes from same individual from different time points

The agreement between the inferred HLA phenotypes with PHLAT from same individual with different time points. Red values indicate the difference between two-digit accuracies between the two different methods while oranges indicate difference between four-digit accuracies.

Hence, we inferred the HLA phenotypes from the 15 patients with T-LGLL profiled with bulk-RNA-seq and from the 9 T-LGLL patients profiled with the scRNA-seq

cohort. The HLA-A*02 genotype, that provided the background for TCRGP models, was the most common in our study as in the Caucasian population, was seen in 6/9 of the T-LGLL samples profiled with scRNA-seq and 10/15 for the samples profiled with bulk-RNA-seq, making 16/24 (66.67%) of the patients HLA-A*02 positive. Additionally, we received HLA information from the proportion of the patients described in the previous publications by Clemente et al. and Kerr et al, and out of which 9/13 (69.23%) and 8/10 (80.00%) T-LGLL clonotypes, respectively, were HLA-A2+.

In total, we were able to confirm the HLA phenotype for 62/199 (31%) T-LGLL clonotypes included in the whole database, out of which 43/62 (69%) were HLA-A*02+. We conclude that 2/199 (1.00%) of T-LGLL clones from HLA-mismatched backgrounds and 0/43 (0%) of T-LGLL clones from HLA-A*02+ do not have an HLA-A*02 antigen target predicted by TCRGP. Conversely, HLA-A*02 negative samples were predicted to also have 0/19 HLA-A*02+ targets, meaning TCRGP did not produce false-positives in this population.

For the TCRGP matches, that were made for 2/199 T-LGLL clonotypes, we were unable to perform HLA genotyping or phenotyping, as one of these TCRs were obtained from the previous publication by Clemente et al¹¹ and the other from our TCR β -seq data set. As we did not have any clone predicted from the HLA-A*02 positive patient, providing a false-negative rate is unfortunately impossible.

We have added the HLA genotyping and HLA-inference to the **Methods** (lines 743-769) and to the **Results** (lines 241-247, 289-300), and **Discussion** (lines 499-507) sections and HLA-genotype information and the predicted HLA-phenotypes to the **Supplementary Table 1** and to the **Supplementary Table 3**.

e. "...the dominant mutated STAT3 clone was replaced by an antigen-driven wild-type STAT3 clone..."

-Is this a leukemic clone with the same TCR α /b? If so, this seems like a very important and interesting finding that should be highlighted more.

Our response: We apologize for the confusion, but the TCR $\alpha\beta$ was not the same. We have rephrased the sentence to state it more clearly.

Nevertheless, we still do think this is an interesting finding, as even though the phenomenon of clonal-drift is known in T-LGLL^{1,11,16}, we are the first to show the phenotype of these drifting clones. For a comparison, we have also provided a case where there is no clonal drift and shown how in this case the phenotype stays stable (**Supplementary Figure 17a-e**).

Minor

1. “Next we analyzed the presence of the 200 T-LGLL clonotypes in a cohort of 785 healthy donors’ TCRb repertoires. Less than half (36.6%) of the clonotypes were found in the healthy TCR repertoires, and they explained <1% of the healthy repertoire.”

- Is this meaningful, given the extreme TCR repertoire heterogeneity between humans who are not HLA-matched?

Our response: Although it is known that T-LGLL TCRs can be found in healthy individuals as well^{10,11}, it has been only thought to appear rarely and our results provide an estimation of how common this phenomenon is (37.19%) in the largest curated database of T-LGLL TCRs.

As stated above, we have now performed HLA inference from our RNA-sequencing data sets and were additionally able to receive the HLA-type for 626/785 (79.75%) healthy individuals from the original publication¹⁷. 294/626 (47.96%) of these healthy donors had a HLA-A*02+ genotype and we could identify 19/43 (44.19%) of the HLA-A*02+ T-LGLL clonotypes in the HLA-A*02+ healthy donors (**Revision Figure 7a-b**).

We have added the **Review Fig 7b** as **Supplementary Fig 10d** and **Supplementary Table 3** and rephrased the **Results** (lines 279-283) accordingly.

Review Figure 7: Public T-LGLL clonotypes in HLA-A*02+ patients and donors

a) Number of times and proportion of TCRs from T-LGLL clonotypes found in healthy donors’ (n=785) TCR repertoire. **b)** The number and percentage of 43 T-LGLL clonotypes from HLA-A*02+ patients and 1+ T-LGLL clonotypes from HLA-A*02 negative patients found in 294 HLA-A*02+ healthy donors

The figure can be seen in **Supplementary Fig. 10b and 10d**.

2. “Non-leukemic T cell populations are more mature and clonal than in other cancers, RA, and healthy controls.”

- Couldn’t this just reflect anti-tumor responses to a blood cancer that is readily seen in lymph nodes and the periphery?

Our response: As discussed above (please see the response to the question 3), unlike in other hematologic malignancies, there is little evidence of anti-tumor response against T-LGLL, and T-LGLL can even be sometimes associated with response to solid or hematologic cancer¹. Hence, we find it unlikely - albeit possible - that what we see is an anti-tumor response. It can be even argued whether T-LGLL is a malignancy or not and should be thought of more as a lymphocytosis. Further, based on the most common symptoms it is closely related to autoimmune disorders, such as rheumatoid arthritis (RA). Hence, we find it interesting, that even when removing the large expanded clones in T-LGLL, the clonality is increased in comparison to RA.

3. Line 331: "...supporting the distinction between the pathogenesis of T-LGLL with mutated and wild-type STAT3."

- The data presented here do not show any difference in pathogenesis as a function of STAT3 mutation.

Our response: We agree with the reviewer's point and have removed this sentence from the **Results** (line 368).

References

1. Lamy, T., Moignet, A. & Loughran, T. P. LGL leukemia: From pathogenesis to treatment. *Blood* **129**, 1082–1094 (2017).
2. Zhang, R. *et al.* Network model of survival signaling in large granular lymphocyte leukemia. *Proceedings of the National Academy of Sciences of the United States of America* **105**, 16308–16313 (2008).
3. Epling-Burnette, P. K. *et al.* Inhibition of STAT3 signaling leads to apoptosis of leukemic large granular lymphocytes and decreased Mcl-1 expression. *The Journal of clinical investigation* **107**, 351–362 (2001).
4. Bilori, B. *et al.* Tofacitinib as a novel salvage therapy for refractory T-cell large granular lymphocytic leukemia. *Leukemia* vol. 29 2427–2429 (2015).
5. Shi, M. *et al.* STAT3 mutation and its clinical and histopathologic correlation in T-cell large granular lymphocytic leukemia. *Human Pathology* **73**, 74–81 (2018).
6. Bandala-Sanchez, E. *et al.* T cell regulation mediated by interaction of soluble CD52 with the inhibitory receptor Siglec-10. *Nature immunology* **14**, 741–748 (2013).
7. Mohan, S. R. *et al.* Therapeutic implications of variable expression of CD52 on clonal cytotoxic T cells in CD8+ large granular lymphocyte leukemia. *Haematologica* **94**, 1407–1414 (2009).
8. Dumitriu, B. *et al.* Alemtuzumab in T-cell large granular lymphocytic leukaemia: interim results from a single-arm, open-label, phase 2 study. *The Lancet. Haematology* **3**, e22–e29 (2016).
9. Mustjoki, S. & Young, N. S. Somatic Mutations in “Benign” Disease. *New England Journal of Medicine* **384**, 2039–2052 (2021).
10. Kerr, C. M. *et al.* Subclonal STAT3 mutations solidify clonal dominance. *Blood Advances* **3**, 917–921 (2019).
11. Clemente, M. J. *et al.* Deep sequencing of the T-cell receptor repertoire in CD8+ T-large granular lymphocyte leukemia identifies signature landscapes. *Blood* (2013) doi:10.1182/blood-2013-05-506386.

12. Messmer, M. *et al.* Large Granular Lymphocytosis With Cytopenias After Allogeneic Blood or Marrow Transplantation: Clinical Characteristics and Response to Immunosuppressive Therapy. *Transplantation and cellular therapy* **27**, 260.e1-260.e6 (2021).
13. Poch Martell, M. *et al.* Distinctive clinical characteristics and favorable outcomes in patients with large granular lymphocytosis after allo-HCT: 12-year follow-up data. *European journal of haematology* **99**, 160–168 (2017).
14. Shugay, M. *et al.* VDJdb: A curated database of T-cell receptor sequences with known antigen specificity. *Nucleic Acids Research* **46**, D419–D427 (2018).
15. Bai, Y., Wang, D. & Fury, W. PHLAT: Inference of High-Resolution HLA Types from RNA and Whole Exome Sequencing BT - HLA Typing: Methods and Protocols. in (ed. Boegel, S.) 193–201 (Springer New York, 2018). doi:10.1007/978-1-4939-8546-3_13.
16. Rajala, H. L. M. *et al.* Discovery of somatic STAT5b mutations in large granular lymphocytic leukemia. *Blood* (2013) doi:10.1182/blood-2012-12-474577.
17. Emerson, R. O. *et al.* Immunosequencing identifies signatures of cytomegalovirus exposure history and HLA-mediated effects on the T cell repertoire. *Nature Genetics* **49**, 659–665 (2017).

REVIEWERS' COMMENTS

Reviewer #2 (Remarks to the Author):

The revised manuscript looks good. No additional requests from this reviewer.

Reviewer #2 (Remarks to the Author):

The authors have adequately addressed the critiques.

Reviewer #3 (Remarks to the Author):

The authors have done an excellent job addressing this reviewer's critiques, especially those relating to clarity and impact of the results presented.

Reviewer #1 (Remarks to the Author):

The revised manuscript looks good. No additional requests from this reviewer.

Our response: We thank the reviewer for taking the time to evaluate our work and the positive feedback.

Reviewer #2 (Remarks to the Author):

The authors have adequately addressed the critiques.

Our response: We thank the reviewer for taking the time to evaluate our work and the positive feedback.

Reviewer #3 (Remarks to the Author):

The authors have done an excellent job addressing this reviewer's critiques, especially those relating to clarity and impact of the results presented.

Our response: We thank the reviewer for taking the time to evaluate our work and the positive feedback.